# POINT-CALIBRATED SPECTRAL NEURAL OPERATORS

## ABSTRACT

Two typical neural models have been extensively studied for operator learning, learning in spatial space via attention mechanism or learning in spectral space via spectral analysis technique such as Fourier Transform. Spatial learning enables point-level flexibility but lacks global continuity constraint, while spectral learning enforces spectral continuity prior but lacks point-wise adaptivity. This work innovatively combines the continuity prior and the point-level flexibility, with the introduced Point-Calibrated Spectral Transform. It achieves this by calibrating the preset spectral eigenfunctions with the predicted point-wise frequency preference via neural gate mechanism. Beyond this, we introduce Point-Calibrated Spectral Neural Operators, which learn operator mappings by approximating functions with the point-level adaptive spectral basis, thereby not only preserving the benefits of spectral prior but also boosting the superior adaptability comparable to the attention mechanism. Comprehensive experiments demonstrate its consistent performance enhancement in extensive PDE solving scenarios.

## 1 INTRODUCTION

Partial differential equations (PDEs) are widely used across a wide range of scientific and engineering tasks, such as airfoil design, plastic structure design, and blood flow simulation. However, traditional PDE solvers depend on high-precision meshes and substantial computational requirements, which significantly impedes efficiency in many engineering applications.

To resolve these limitations, recent works (Li et al., 2020; Lu et al., 2019; Tripura & Chakraborty, 2022) introduce neural operators, a class of data-driven approaches that directly learn mappings between continuous function spaces for solving parametric partial differential equations. The most performed neural operators could be classified into two groups, i.e., attention-based neural operators (Cao, 2021; Hao et al., 2023; Wu et al., 2024) and spectral-based neural operators (Li et al., 2020; Tran et al., 2021; Gupta et al., 2021). Attention-based methods directly learn operators in the original physical space, devoid of prior constraints. In contrast, spectral-based methods learn operators in a truncated spectral space via spectral transformation such as Fourier Transform, seamlessly integrating the resolution-invariant prior.

Both attention-based neural operators and spectral-based neural operators exhibit distinct advantages and limitations. **Attention-based neural operators** (Hao et al., 2023; Xiao et al., 2023; Wu et al., 2024) can adapt to various physical systems and obtain consistent leading performance on problems with sufficient training data amount, benefiting from the flexible spatially point-wise learning on the physical domain. However, the pure data-driven framework without spectral prior limits their generalization capability, thus suffering serious performance drops in scenarios with scarce data, which is common in practical applications. In contrast, **Spectral-based neural operators** (Li et al., 2020; Kovachki et al., 2023; Tran et al., 2021) can efficiently learn operator mappings between continuous spaces with limited training data, through approximating physical functions in the truncated spectral space. However, the classical spectral processing mechanism lacks point-level flexibility for adaptively handling the spatially varying phenomenon in physical systems. This makes them struggle to resolve complex PDEs and constrains their performance promotion with the increasing of training data amount. Therefore, both attention-based neural operators and existing spectral-based neural operators struggle to manage various PDE solving scenarios.

This work aims to develop more advanced neural operators that not only have strong generalization capability like previous spectral-based methods (Li et al., 2020; Tran et al., 2021) but also possess point-level flexibility like the attention-based methods (Hao et al., 2023; Wu et al., 2024). We present

**Point-Calibrated Spectral Transform**, an improved spectral transform technique that integrates point-wise frequency preference for flexible spectral feature learning. Specifically, we first predict the frequency preference of each physical point via neural gate mechanism, and then the frequency preference is used to calibrate spectral eigenfunctions. This enables feature learning in a point-adaptive spectral space where spatially point-wise status is integrated, rather than point-irrelevant spectral space like previous approaches (Li et al., 2020; Tran et al., 2021). Next, we introduce **Point-Calibrated Spectral Mixer based Neural Operator** (simply denoted as PCSM), where operator mappings are learned by approximating functions with adaptive spectral basis based on Point-Calibrated Spectral Transform. In PCSM, the spectral prior enables efficient operator learning even under scarce training data amount, and the point-level calibration enables adaptively handling spatially varying phenomena e.g. adding high-frequency features in regions with sharp status changes.

Extensive experiments are conducted to validate the superiority of Point-Calibrated Spectral Mixer. **(a)** First, we compare PCSM with previous most performed neural operators on diverse PDE solving problems, demonstrating its leading performance over both existing spectral-based methods and attention-based methods. **(b)** Additionally, our experimental results validate that PCSM simultaneously has the advantages of spectral-based and attention-based neural operators. Similar to spectral-based methods, PCSM can be efficiently optimized during training, performs well under limited training data, and possesses great zero-shot resolution generalization capability. Similar to attention-based methods, PCSM can flexibly manage different PDE problems and continuously achieves significant performance gains as training data amount increases. And we find that PCSM performs well even under extremely little spectral frequencies, different from previous spectral-based methods that rely on sufficient spectral frequencies. **(c)** Furthermore, visualization analysis of the learned frequency preference is provided. We find that the learned frequency preference by PCSM can instruct the frequency design for constructing better fixed spectral-based neural operators.

Our core contributions are summarized as follows:

- We present Point-Calibrated Spectral Transform, pioneeringly learning features in a point-status integrated spectral space, holding potential applications in diverse spectrum-related tasks.
- We present Point-Calibrated Spectral Neural Operator (PCSM), which **(a)** performs well under limited training data and unseen resolutions like spectral-based methods, **(b)** flexibly handles various PDEs and efficiently utilizes training data like attention-based methods.
- We demonstrate the superiority of PCSM in various scenarios through comprehensive experiments, and find the frequency preference learned by PCSM can help design spectral-based neural operators.

## 2 METHODOLOGY

### 2.1 PRELIMINARY

#### 2.1.1 PROBLEM FORMULATION

Following previous works (Li et al., 2020; Lu et al., 2021; Kovachki et al., 2023), we formulate the solution of parametric partial differential equations as the operator mapping between two infinite-dimensional function spaces:

$$\mathcal{G}^{\dagger} : \mathcal{A} \to \mathcal{U}, \mathcal{A} = \{\boldsymbol{a} | \boldsymbol{a} : \Omega \to \mathbb{R}^{d_a}\}, \mathcal{U} = \{\boldsymbol{u} | \boldsymbol{u} : \Omega \to \mathbb{R}^{d_u}\}, \tag{1}$$

where $\Omega$ denotes the physical domain, $d_a$ and $d_u$ represent the channel number of input functions and output functions respectively. The function $\boldsymbol{a}$ and $\boldsymbol{u}$ are the state functions defined on the problem domain. They are differently instantiated for different PDE problems. For example, in the steady-state problem Darcy Flow, $\boldsymbol{a}$ denotes the diffusion coefficient and $\boldsymbol{u}$ represents the solution function. In the time-series problem Navier-Stokes, $\boldsymbol{a}$ is the vorticity states in previous time steps and $\boldsymbol{u}$ is the vorticity states of following time steps. The operator learning problem is to learn a parameterized surrogate model $\mathcal{G}_{\boldsymbol{\theta}}^{\dagger}$ for the operator mapping $\mathcal{G}^{\dagger}$. Specifically, we need to train a neural operator network $\mathcal{G}_{\boldsymbol{\theta}}^{\dagger}$, which takes $(\boldsymbol{a}, \boldsymbol{g})$ as input and produces the output function $\boldsymbol{u}$.

#### 2.1.2 TRANSFORMER-BASED NEURAL OPERATOR

Transformer (Vaswani, 2017) has been a typical choice for neural operator learning (Tran et al., 2021; Hao et al., 2023; Wu et al., 2024). First, an element-wise projecting layer $P$ maps the input function

$\boldsymbol{a}$ to a latent function $\boldsymbol{v}_0 \in \mathbb{R}^{N \times d_v}$ in a high-dimensional space, where $d_v$ denotes the number of latent dimensions. Then, the latent function passes through a stack of feature mixing modules $(M_1, M_2, ..., M_l)$, where $l$ represents the network depth. Each $M_i$ takes $\boldsymbol{v}_{i-1} \in \mathbb{R}^{N \times d_v}$ as input and produces $\boldsymbol{v}_i \in \mathbb{R}^{N \times d_v}$ as output. Finally, a mapping layer $V$ transforms the last hidden function $\boldsymbol{v}_l$ to the target function $\boldsymbol{u} \in \mathbb{R}^{N \times d_u}$. This process can be represented by the equation below:

$$\mathcal{G}_\theta^\dagger = V \circ M_l \circ ... \circ M_2 \circ M_1 \circ P, \tag{2}$$

where $P$ and $V$ are typically implemented using fully connected layers. Each feature mixing block $M_i$ is non-local learnable neural operators (Kovachki et al., 2023), as detailed in Section. A.2.2. In Transformer-based models, $M_i$ consists of token mixing and channel mixing blocks:

$$\text{Token Mixing} : \boldsymbol{v}_{i-1}^{\text{mid}} = \mathcal{F}^{\text{mixer}}(\text{LayerNorm}(\boldsymbol{v}_{i-1})) + \boldsymbol{v}_{i-1}, \tag{3}$$

$$\text{Channel Mixing} : \boldsymbol{v}_i = \text{FeedForward}(\text{LayerNorm}(\boldsymbol{v}_{i-1}^{\text{mid}})) + \boldsymbol{v}_{i-1}^{\text{mid}}, \tag{4}$$

where $\text{LayerNorm}(\cdot)$ and $\text{FeedForward}(\cdot)$ are the layer normalization and feed-forward layer, respectively. $\mathcal{F}^{\text{mixer}}$ represents the operation for information mixing along the spatial dimension such as convolution and self-attention. For example, recent works (Cao, 2021; Hao et al., 2023; Wu et al., 2024) explore attention-based token mixers for operator learning, demonstrating the versatility of this approach in capturing complex spatial relationships. Though the attention-based token mixer possesses enough flexibility to adapt to various physical systems, the lack of prior constraints makes it suffer serious performance drops under limited training data and unseen low-resolution samples, as shown in Table. 3, 4.

**Spectral-based Token Mixer.** Spectral neural operators offer an alternative approach to capturing complex relationships in operator learning. These operators leverage spectral transforms to capture and manipulate frequency-domain representations of the input data. The typical spectral transform methods employed include the Discrete Fourier Transform in FNO (Li et al., 2020) and Wavelet Transform in WNO (Tripura & Chakraborty, 2022). The spectral-based token mixer can be formulated as follows:

$$\mathcal{F}_{\text{spectral}}^{\text{mixer}}(\boldsymbol{x}) = \mathcal{T}^{-1} \circ \text{Project} \circ \mathcal{T}(\boldsymbol{x}), \tag{5}$$

where $\mathcal{T}(\cdot)$ represents the spectral transform operator, yielding spectral feature $\hat{\boldsymbol{x}} \in \mathbb{R}^{N^k \times d_v}$. $N^k$ represents the number of retained frequencies in the spectral domain. $\mathcal{T}^{-1}(\cdot)$ denotes the inverse spectral transform operator, producing spatial feature $\boldsymbol{x} \in \mathbb{R}^{N \times d_v}$. $\text{Project}(\cdot)$, typically implemented as a fully connected layer, is a transformation designed to capture complex relationships in the frequency domain. By operating in the frequency domain, spectral neural operators can efficiently learn patterns that may be less apparent in the spatial domain, potentially mitigating the performance degradation observed in attention-based token mixers under data constraints.

## 2.2 POINT-CALIBRATED SPECTRAL NEURAL OPERATOR

In this section, we propose a new transformer-based neural operator that achieves superior performance across various scenarios by integrating the point-level flexibility of attention-based mixers with the spectral prior of spectral-based mixers.

### 2.2.1 LAPLACE-BELTRAMI TRANSFORM

While coordinate-based spectral transforms such as Fourier Transform and Wavelet Transform are prevalent in existing neural operators (Li et al., 2020; Tran et al., 2021; Gupta et al., 2021), they are limited by uniform discretization requirements and applicability to regular domains. To address these constraints and enable spectral processing on general physical domains, we introduce the Laplace-Beltrami Transform, following Chen et al. (2023). This approach adapts to irregular geometries and non-uniform meshes, potentially enhancing the versatility and accuracy of neural operators across diverse physical scenarios.

**Laplace-Beltrami Transform.** For spectral processing on the general physical domain, we follow Chen et al. (2023) and utilize Laplace-Beltrami Operator (LBO) eigenfunctions (Rustamov et al., 2007) to transform the features between the spectral and physical domain. LBO eigenfunctions constitute a set of spectral basis (Patanè, 2018) of manifolds, which have been demonstrated as the optimal basis for function approximating on Riemannian manifold (Aflalo et al., 2015). The LBO eigenfunctions can be represented as a list of functions $\phi_i \in \mathbb{R}^{N \times 1}$, where each $\phi_i$ is an eigenfunction

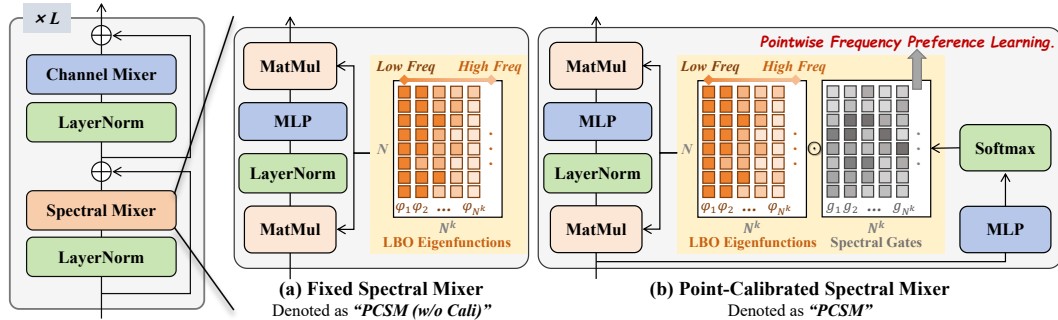

Figure 1: The overall architecture of Point-Calibrated Spectral Neural Operator.

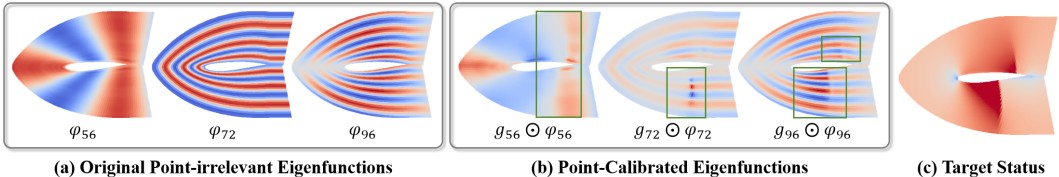

Figure 2: Visualization of Point-Calibrated spectral eigenfunctions on Airfoil. The green boxes signify the calibrated regions with point-level frequency preference.

of the Laplace-Beltrami operator on the manifold. These eigenfunctions have correspondence to spectral frequencies: eigenfunctions associated with lower eigenvalues correspond to lower spectral frequencies, while those associated with larger eigenvalues correspond to higher spectral frequencies.

Consider the latent function $\boldsymbol{x} \in \mathbb{R}^{N \times d_v}$, the matrix form of spectral transform $\mathcal{T}_{\text{LBT}}$ and inverse transform $\mathcal{T}_{\text{LBT}}^{-1}$ could be formulated as follows:

$$\mathcal{T}_{\text{LBT}}(\boldsymbol{x}) = [\boldsymbol{x}^T \phi_1, \boldsymbol{x}^T \phi_2, ..., \boldsymbol{x}^T \phi_{N^k}]^T, \tag{6}$$

$$\mathcal{T}_{\text{LBT}}^{-1}(\hat{\boldsymbol{x}}) = [\phi_1, \phi_2, ..., \phi_{N^k}]\hat{\boldsymbol{x}}, \tag{7}$$

where $\boldsymbol{x}^T \phi_i$ represents the $i$-th frequency feature of $\boldsymbol{x}$. $\hat{\boldsymbol{x}} \in \mathbb{R}^{N^k \times d_v}$ is the truncated spectrum of $\boldsymbol{x}$. $N^k$ is the number of remained frequencies after truncation. To facilitate understanding, we also provide the formal definition of the transformations in Section. A.2.1. In this work, we use the robust-laplacian (Sharp & Crane, 2020) library [1] to calculate the LBO eigenfunctions for specific physical domain. It allows calculating the laplacian for triangle meshes or point clouds of general physical domain. Thus, we can learn neural operators on both structured domains (using laplacian of points from manually constructed regular grids) and unstructured domains (using laplacian of irregular meshes). Additionally, for handling some complex PDEs, we manually add high-frequency spectrum with Sparse-Frequency Spectral Transform, as shown in Section. A.1.2.

### 2.2.2 POINT-CALIBRATED LAPLACE-BELTRAMI TRANSFORM

We propose the Point-Calibrated Laplace-Beltrami Transform to enhance spectral neural operators by combining the spatial learning of attention-based mixers with the strong priors of spectral-based mixers. By integrating Laplace-Beltrami Operator (LBO) eigenfunctions with point-wise calibration, we allow frequency selection to vary across physical points. This mechanism can effectively capture spatially varying phenomena in PDE systems. Our method maintains the computational efficiency and generalization strengths of spectral approaches while incorporating the spatial adaptability of attention-based methods, boosting performance across diverse PDE problems.

Given an input feature $x \in \mathbb{R}^{N \times d_v}$, we predict point-wise spectral gates $\boldsymbol{G} \in [0,1]^{N \times N^k}$ by an MLP layer followed by a Softmax function:

$$\boldsymbol{G} = [\boldsymbol{g}_1, \boldsymbol{g}_2, ..., \boldsymbol{g}_{N^k}] = \text{Softmax}(\text{MLP}_{\text{gate}}^{N^k}(x)), \tag{8}$$

where $N^k$ indicates the number of output frequencies. Softmax$(\cdot)$ is applied along the channel dimension and produces normalized gates for each physical point. $\boldsymbol{g}_i \in [0,1]^{N \times 1}$ represents the

---

[1]Robust-laplacian library link: https://github.com/nmwsharp/nonmanifold-laplacian

frequency preference of each physical point for the $i$-th frequency, allowing for point-wise spectral modulation. We implement the point-calibrated transform using point-wise spectral gates $\boldsymbol{G}$:

$$\mathcal{T}_{\text{PC-LBT}}(\boldsymbol{x}) = [\boldsymbol{x}^T(\boldsymbol{g}_1 \odot \phi_1), \boldsymbol{x}^T(\boldsymbol{g}_2 \odot \phi_2), ..., \boldsymbol{x}^T(\boldsymbol{g}_{N^k} \odot \phi_{N^k})]^T, \tag{9}$$

$$\mathcal{T}_{\text{PC-LBT}}^{-1}(\hat{\boldsymbol{x}}) = [\boldsymbol{g}_1 \odot \phi_1, \boldsymbol{g}_2 \odot \phi_2, ..., \boldsymbol{g}_{N^k} \odot \phi_{N^k}]\hat{\boldsymbol{x}}, \tag{10}$$

where $g_i \odot \phi_i$ denotes the calibrated eigenvector for the $i$-th frequency. $\boldsymbol{x}$ and $\hat{\boldsymbol{x}}$ denote the original features and transformed features, respectively. We demonstrate the spectral mixer based on $\mathcal{T}_{\text{PC-LBT}}$ and $\mathcal{T}_{\text{PC-LBT}}^{-1}$ is an integral operator in Theorem. A.8. It essentially integrates the integral kernels of fixed spectral transform and the linear attention mechanism, as detailed in Remark. A.9.

As shown in Figure. 2, the point-wise calibrated eigenvector modulates $\phi_i$ individually for each point, integrating varied frequency preferences across physical points. Unlike the standard Laplace-Beltrami Transform, where each point's importance is tied solely to its geometric location, our approach considers both location and physical state. This allows emphasis on points experiencing significant physical changes and enables different spectral modulations for latent features across layers.

**Discussion.** Prior adaptive frequency selection methods (Guibas et al., 2021; George et al., 2022; Li & Yang, 2023) attempt to directly select frequencies in the spectral domain, resulting in a shared frequency filter for all points, labeled as *global-level frequency selection*. Conversely, we cultivate spectral gates in the spatial domain and utilize the gates during spectral transformation, allowing each point to select its own suited frequencies, labeled as *point-level frequency selection*.

Point-Calibrated Spectral Transform enjoys three advantages: **(a)** PCSM automatically determines appropriate frequency ranges and combinations for various PDE problems, adapting to domain geometry and operator types without manual spectrum design. This allows flexible frequency assignment to each point based on both location and physical state. **(b)** PCSM combines the efficient convergence and strong generalization of spectral-based methods with the scalability of attention-based approaches. This fusion enables effective performance across diverse PDE problems and data scales. **(c)** The learned spectral gates reflect the frequency preference of physical points, which can be used for additional applications such as guiding spectral design for fixed spectral models.

### 2.2.3 POINT-CALIBRATED SPECTRAL NEURAL OPERATOR

We introduce the Point-Calibrated Spectral Neural Operator, a transformer-based architecture that integrates Point-Calibrated Spectral Transform with multi-head processing for enhanced performance in modeling complex physical systems.

Similar to multi-head self-attention (Vaswani, 2017), we enhance the spectral mixer by introducing the multi-head spectral mixer. Specifically, we first split the latent features $\boldsymbol{x} \in \mathbb{R}^{N \times d_v}$ into $h$ vectors $\boldsymbol{x}^{\text{head-1}}, \boldsymbol{x}^{\text{head-2}}, ..., \boldsymbol{x}^{\text{head-h}}$ along the channel dimension, where $\boldsymbol{x}^{\text{head-i}} = \boldsymbol{x}_{[:, d_v^{\text{head}} \times (i-1):d_v^{\text{head}} \times i]}$ and $h$ denotes the number of heads. $d_v^{\text{head}} = d_v/h$ is the dimension of features in single head. Next, every vector $\boldsymbol{x}^{\text{head-i}} \in \mathbb{R}^{N \times d_v^{\text{head}}}$ is independently processed by $\mathcal{F}_{\text{spectral}}^{\text{mixer}}$. Finally, all vectors are concatenated as the output. The multi-head mixer could be formulated as follows:

$$\mathcal{F}_{\text{spectral}}^{\text{mixer}}(\boldsymbol{x}) = \mathcal{T}_{\text{PC-LBT}}^{-1} \circ \text{FC} \circ \text{LayerNorm} \circ \mathcal{T}_{\text{PC-LBT}}(\boldsymbol{x}), \tag{11}$$

$$\mathcal{F}_{\text{spectral}}^{\text{multi-head-mixer}}(\boldsymbol{x}) = \text{Concat}(\mathcal{F}_{\text{spectral}}^{\text{mixer}}(\boldsymbol{x}^{\text{head-i}})). \tag{12}$$

LayerNorm$(\cdot)$ is introduced to normalize the spectral features for more efficient optimization and enhanced generalization. Additionally, we share the learnable weights of FC for all spectrum frequencies. By enabling point-wise frequency modulation, our approach offers enhanced modeling capabilities for a wide range of complex PDE problems across various physical domains.

## 3 EXPERIMENT

We evaluate PCSM with extensive experiments, including structured mesh problem solving in Section. 3.1, unstructured mesh problem solving in Section. 3.2, generalization capability evaluation in Section. 3.3, point-wise frequency preference analysis in Section. 3.4, and ablations in Section. A.4.

### 3.1 STRUCTURED MESH PROBLEMS

This section compares PCSM with previous neural operators on structured mesh problems, where the physical domains are represented with meshes aligned with standard rectangle grids. For these problems, we implement PCSM with LBO eigenfunctions calculated on standard rectangle grids.

Table 1: Performance comparison on structured mesh problems.

| Model | Darcy Flow (Regular, Steady) | Airfoil (Irregular, Steady) | Navier-Stokes (Regular, Time) | Plasticity (Irregular, Time) |
|---|---|---|---|---|
| FNO (Li et al., 2020) | 1.08e-2 | - | 1.56e-1 | - |
| WMT (Gupta et al., 2021) | 8.20e-3 | 7.50e-3 | 1.54e-1 | 7.60e-3 |
| U-FNO (Wen et al., 2022) | 1.83e-2 | 2.69e-2 | 2.23e-1 | 3.90e-3 |
| Geo-FNO (Li et al., 2023c) | 1.08e-2 | 1.38e-2 | 1.56e-1 | 7.40e-3 |
| U-NO (Rahman et al., 2022) | 1.13e-2 | 7.80e-3 | 1.71e-1 | 3.40e-3 |
| F-FNO (Tran et al., 2021) | 7.70e-3 | 7.80e-3 | 2.32e-1 | 4.70e-3 |
| LSM (Wu et al., 2023) | 6.50e-3 | 5.90e-3 | 1.54e-1 | 2.50e-3 |
| NORM (Chen et al., 2023) | 9.71e-3 | 5.44e-3 | 1.15e-1 | 4.39e-3 |
| Galerkin (Cao, 2021) | 8.40e-3 | 1.18e-2 | 1.40e-1 | 1.20e-2 |
| HT-Net (Liu et al., 2023) | 7.90e-3 | 6.50e-3 | 1.85e-1 | 3.33e-2 |
| OFORMER (Li et al., 2023a) | 1.24e-2 | 1.83e-2 | 1.71e-1 | 1.70e-3 |
| GNOT (Hao et al., 2023) | 1.05e-2 | 7.60e-3 | 1.38e-1 | 3.36e-2 |
| FactFormer (Li et al., 2023b) | 1.09e-2 | 7.10e-3 | 1.21e-1 | 3.12e-2 |
| ONO (Xiao et al., 2023) | 7.60e-3 | 6.10e-3 | 1.20e-1 | 4.80e-3 |
| Transolver (Wu et al., 2024) | 5.70e-3$_{\pm 1.00e\text{-}4}$ | 5.30e-3$_{\pm 1.00e\text{-}4}$ | 9.00e-2$_{\pm 1.30e\text{-}3}$ | 1.23e-3$_{\pm 1.00e\text{-}4}$ |
| PCSM (w/o Cali) | 5.41e-3$_{\pm 1.15e\text{-}4}$ | 5.13e-3$_{\pm 1.75e\text{-}4}$ | 9.46e-2$_{\pm 1.24e\text{-}3}$ | 1.21e-3$_{\pm 3.61e\text{-}5}$ |
| PCSM | **4.59e-3**$_{\pm 1.94e\text{-}4}$ | **4.72e-3**$_{\pm 1.51e\text{-}4}$ | **7.34e-2**$_{\pm 9.22e\text{-}4}$ | **8.00e-4**$_{\pm 4.58e\text{-}5}$ |

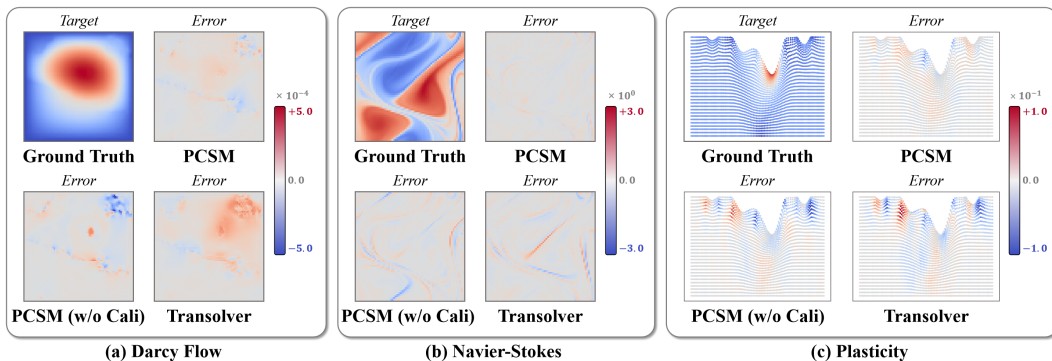

(a) Darcy Flow  (b) Navier-Stokes  (c) Plasticity

Figure 3: Prediction error visualization on different problems.

**Setup.** **(a)** Problems. The experimental problems include two regular domain problems Darcy Flow and Navier-Stokes from Li et al. (2020), and two irregular domain problems Airfoil and Plasticity from Li et al. (2023c). Darcy Flow and Airfoil are steady-state solving problems, while Navier-Stokes and Plasticity are time-series solving problems. **(b)** Metric. Same as previous works (Li et al., 2020), we use Relative L2 between the predicted results and ground truth (the simulated results) as the evaluation metric, lower value indicating higher PDE solving accuracy. **(c)** Baselines. We compare PCSM with a lot of neural operators, covering both spectral-based methods and attention-based methods. Section. A.3 presents more experimental setup detail.

**Quantitative Comparison.** Table. 1 presents the quantitative results. PCSM significantly improves the performance over past spectral-based methods LSM (Wu et al., 2023) and NORM (Chen et al., 2023), and outperforms the most performed attention-based method Transolver (Wu et al., 2024). This concludes that calibrated spectral basis effectively resolves the inflexibility of spectral-based methods and learns better features for operator learning on various problems.

**Qualitative Comparison.** In Figure. 3, we visualize the prediction error of PCSM and Transolver on different problems. Compared to Transolver (Wu et al., 2024), the prediction error is evidently reduced, especially on physical boundaries and some regions with sharp status changes. This further demonstrates the superior operator learning capability of PCSM.

**Optimization Efficiency Comparison.** In addition, we compare the validation loss curves of PCSM and Transolver during training, as portrayed in Figure 4. We notice that PCSM reaches the same prediction accuracy as Transolver earlier, often dozens or even hundreds of epochs ahead, especially in the initial and middle stages of training (the first 300 epochs). This confirms the excellent operator

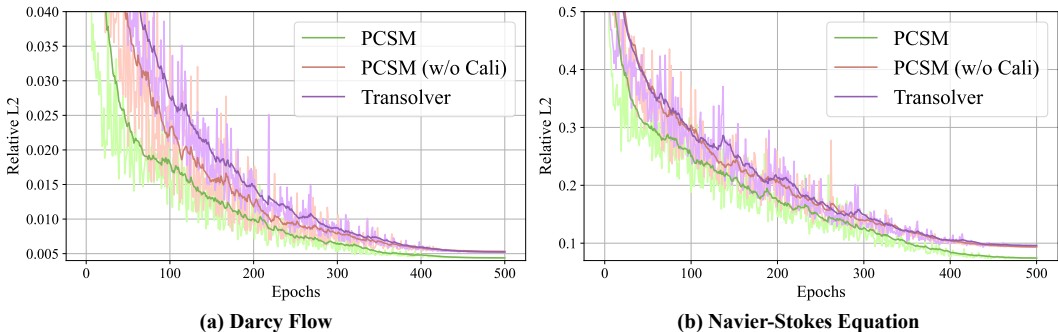

(a) Darcy Flow
(b) Navier-Stokes Equation

Figure 4: Comparison of validation loss curve during training.

Table 2: Performance comparison on unstructured mesh problems.

| Model | Irregular Darcy (2290 Nodes) | Pipe Turbulence (2673 Nodes) | Heat Transfer (7199 Nodes) | Composite (8232 Nodes) | Blood Flow (1656 Nodes) |
|---|---|---|---|---|---|
| GraphSAGE (Hamilton et al., 2017) | 6.73e-2$_{\pm5.30e\text{-}4}$ | 2.36e-1$_{\pm1.41e\text{-}2}$ | - | 2.09e-1$_{\pm5.00e\text{-}4}$ | - |
| DeepOnet (Lu et al., 2019) | 1.36e-2$_{\pm1.30e\text{-}4}$ | 9.36e-2$_{\pm1.07e\text{-}3}$ | 7.20e-4$_{\pm2.00e\text{-}5}$ | 1.88e-2$_{\pm3.40e\text{-}4}$ | 8.93e-1$_{\pm2.37e\text{-}2}$ |
| POD-DeepOnet (Lu et al., 2022) | 1.30e-2$_{\pm2.30e\text{-}4}$ | 2.59e-2$_{\pm2.75e\text{-}3}$ | 5.70e-4$_{\pm1.00e\text{-}5}$ | 1.44e-2$_{\pm6.00e\text{-}4}$ | 3.74e-1$_{\pm1.19e\text{-}3}$ |
| FNO (Li et al., 2020) | 3.83e-2$_{\pm7.70e\text{-}4}$ | 3.80e-2$_{\pm2.00e\text{-}5}$ | - | - | - |
| NORM (Chen et al., 2023) | 1.05e-2$_{\pm2.00e\text{-}4}$ | 1.01e-2$_{\pm2.00e\text{-}4}$ | 2.70e-4$_{\pm2.00e\text{-}5}$ | 9.99e-3$_{\pm2.70e\text{-}4}$ | 4.82e-2$_{\pm6.10e\text{-}4}$ |
| PCSM (w/o Cali) | 7.96e-3$_{\pm7.19e\text{-}5}$ | 1.11e-2$_{\pm1.00e\text{-}3}$ | 1.11e-3$_{\pm3.25e\text{-}4}$ | 1.00e-2$_{\pm5.24e\text{-}4}$ | 3.73e-2$_{\pm5.83e\text{-}4}$ |
| PCSM | **7.38e-3**$_{\pm6.20e\text{-}5}$ | **8.26e-3**$_{\pm7.60e\text{-}4}$ | **1.84e-4**$_{\pm2.27e\text{-}5}$ | **9.34e-3**$_{\pm2.71e\text{-}4}$ | **2.89e-2**$_{\pm3.25e\text{-}3}$ |

fitting ability of PCSM benefiting from the suitable combination of a spectral continuity prior (offering fundamental function approximation basis) and point-wise calibration (providing efficient adaptivity).

## 3.2 Unstructured Mesh Problems

This section compares PCSM with previous works on unstructured mesh problems, where the physical domains are represented with irregular triangle meshes. For handling these problems, we independently calculate LBO eigenfunctions for each problem based on their triangle meshes.

**Setup.** **(a)** Problems. The evaluated problems include Irregular Darcy, Pipe Turbulence, Heat Transfer, Composite, and Blood Flow from Chen et al. (2023). All problems come from realistic industry scenarios and include both steady-state problems and time-series problems. **(b)** Metric. Same as Section. 3.1, Relative L2 between the predicted results and ground truth (the simulated results) is used as the evaluation metric, lower value indicating better performance. **(c)** Baselines. The compared methods include GraphSAGE (Hamilton et al., 2017), DeepOnet (Lu et al., 2019), POD-DeepOnet (Lu et al., 2022), FNO (Li et al., 2020) and NORM (Chen et al., 2023). Section. A.3 presents more experimental setup detail.

**Results.** The results are shown in Table. 2. Compared to previous methods, PCSM obtains consistent enhanced performance across all problems. This validates the benefits of point-adaptive spectral feature learning of PCSM on complex physical domains and operator mappings.

## 3.3 Generalization Capability Comparison

This section compares the generalization performance of PCSM and the best attention-based neural operator Transolver (Wu et al., 2024), including zero-shot testing on unseen sample resolutions and neural operator learning on limited training data amount.

**Zero-shot Resolution Generalization.** We evaluate the zero-shot capabilities of PCSM and Transolver on samples with unseen resolutions, on the irregular domain problem, Airfoil. The model is trained on the $211 \times 51$ resolution and then tested on lower

Table 3: Zero-shot resolution generalization on Airfoil.

| Resolution Type | Test Resolution | Transolver (Wu et al., 2024) | PCSM (w/o Cali) | PCSM (w/ Cali) |
|---|---|---|---|---|
| Original Ratio | $111 \times 26$ | 7.68e-2 | 1.90e-2 | **1.74e-2** |
| | $45 \times 11$ | 9.73e-2 | 7.30e-2 | **5.34e-2** |
| Varied Ratio | $221 \times 26$ | 7.85e-2 | 1.91e-2 | **1.69e-2** |
| | $111 \times 51$ | 1.26e-2 | 5.80e-3 | **5.37e-3** |

resolutions including $111 \times 26$ and $45 \times 11$, as well as varied ratio resolutions including $221 \times 26$ and

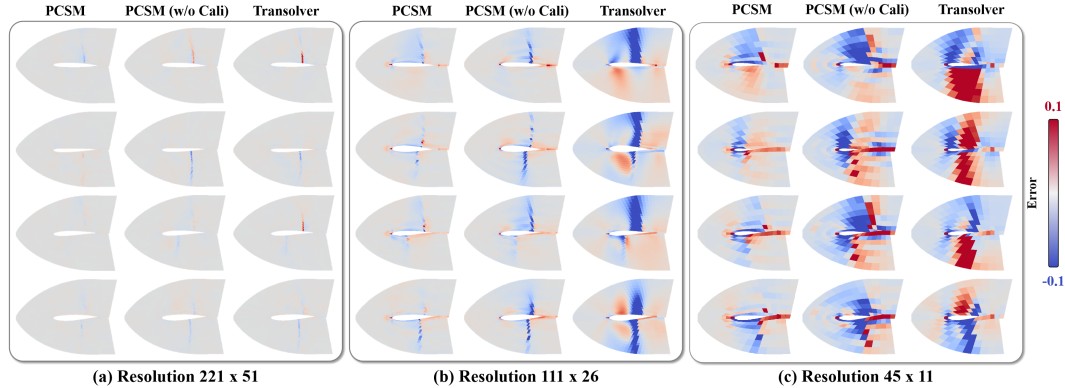

PCSM   PCSM (w/o Cali)   Transolver   PCSM   PCSM (w/o Cali)   Transolver   PCSM   PCSM (w/o Cali)   Transolver

(a) Resolution 221 x 51    (b) Resolution 111 x 26    (c) Resolution 45 x 11

Figure 5: Visualization of prediction error on different test resolutions.

$111 \times 51$. We utilize Relative L2 as the performance metric, with a lower value indicating preferred performance.

Table. 3 presents the quantitative comparison results, where significant performance gaps between Transolver and PCSM are observed. Additionally, we visualize the prediction error of different resolutions in Figure. 5. In contrast to Transolver, PCSM significantly diminishes prediction error, particularly on lower-resolution samples. Our results show that PCSM retains its remarkable resolution generalization ability akin to previous spectral-based neural operators (Li et al., 2020), different from the purely attention-based method (Wu et al., 2024) that encounters performance declines on unseen resolutions despite its superior flexibility.

**Limited Training Numbers.** We additionally evaluate the generalization ability of PCSM and Transolver (Wu et al., 2024) with limited training data amount. Specifically, for Darcy Flow and Navier-Stokes, we train neural operators with 200, 400, 600, 800, and 1000 trajectories respectively, and then tested on extra 200 trajectories. We use Relative L2 as the performance measure, with a lower value meaning better performance.

Table 4: Comparison on different training numbers.

| Problem | Training Number | Transolver (Wu et al., 2024) | PCSM (w/o Cali) | PCSM (w/ Cali) |
|---------|-----------------|------------------------------|-----------------|----------------|
| Darcy Flow | 200 | 1.75e-2 | 1.10e-2 | **1.06e-2** |
| | 400 | 1.04e-2 | 7.32e-3 | **6.66e-3** |
| | 600 | 6.87e-3 | 6.20e-3 | **6.03e-3** |
| | 800 | 6.33e-3 | 5.64e-3 | **4.98e-3** |
| | 1000 | 5.24e-3 | 5.33e-3 | **4.38e-3** |
| Navier-Stokes | 200 | 3.76e-1 | 1.93e-1 | **1.85e-1** |
| | 400 | 3.14e-1 | 1.48e-1 | **1.26e-1** |
| | 600 | 2.87e-1 | 1.21e-1 | **1.17e-1** |
| | 800 | 2.49e-1 | 1.04e-1 | **8.25e-2** |
| | 1000 | 9.60e-2 | 9.34e-2 | **7.44e-2** |

Table. 4 reports the results. Overall, PCSM consistently outperforms Transolver across all quantities of training data. When there is little training data, PCSM is as good as the fixed spectral baseline (called "PCSM w/o Cali") and significantly outperforms the attention-based Transolver method. As the training data amount increases, PCSM efficiently utilizes available data like Transolver, obtaining consistent performance leading. This illustrates that PCSM maintains the spectral basis prior while gaining flexibility, thus accounting for its superior performance under various circumstances.

### 3.4    ANALYSIS OF POINT-WISE FREQUENCY PREFERENCE LEARNED BY PCSM

In this section, we study the frequency augmentation mechanism learned by PCSM, exploring where and at what level PCSM enhances high or low frequency spectral basis for each physical point.

**Point-wise Frequency Preference Visualization.** In Figure. 6, the spectral gates $G$ of a 4-layer PCSM for Darcy Flow and an 8-layer PCSM for Airfoil are displayed. Each physical point's spectral gates $G$ (comprising $N^k$ gate values associated with different frequencies for each point) are visualized by calculating the difference between the sum of high-frequency gates (the last $N^k/2$ values) and low-frequency gates (the first $N^k/2$ values). The outcome is a frequency intensity value ranging between $[-1, 1]$, with larger values (red color) indicating preferring high frequencies and smaller values (blue color) indicating preferring low frequencies.

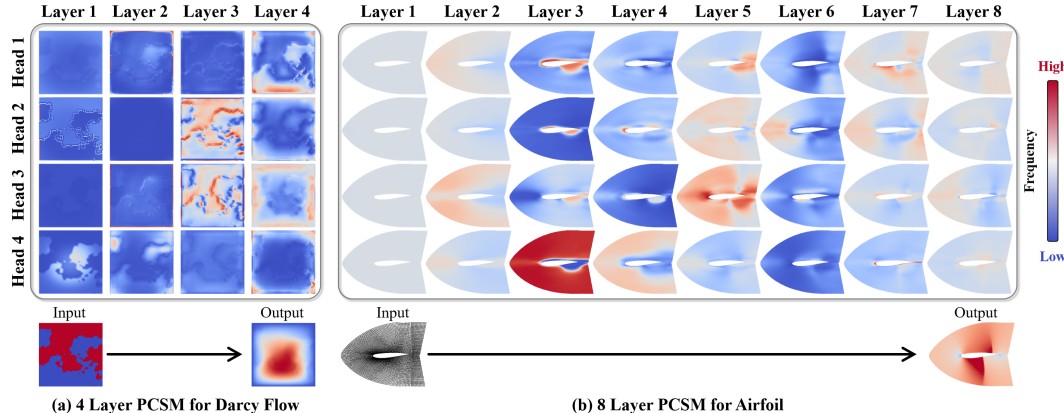

**Figure 6:** Visualization of point-wise frequency preference on Darcy Flow and Airfoil.

Figure. 6 shows the learned frequency preference of each physical point in different layers and heads. More visualization results of different resolution samples and different sample cases are presented in Figure. 7 and Figure. 8 respectively. The following empirical results are observed: **(a)** Generally, it prioritizes low-frequency basis for neural operator learning, which aligns with earlier studies (Li et al., 2020; Chen et al., 2023) favoring the lowest frequencies. **(b)** Boundary regions and rapidly physical quantity changing areas lean towards taking more high-frequency spectral basis. This suggests their need for detailed information. **(c)** The primary enhancement of high-frequency spectrum appears in the neural network's middle layers, diminishes in the late layers, and is nearly absent in the early layers.

**Fixed Frequency Design Guided by Spectral Gates.** As previously demonstrated, notable performance enhancements have been achieved in extensive problems via the point-adaptive spectral calibration by PCSM. This raises the question: is the layer selection strategy for frequency calibration learned from data universally beneficial?

**Table 5:** Manually fixed spectral design guided by spectral gates of PCSM on Darcy Flow.

| Spectral Types | Relative Error |
|---|---|
| Manually Add High Frequency in Early Layers | 5.12e-3 |
| Manually Add High Frequency in Middle Layers | 4.84e-3 |
| Manually Add High Frequency in Late Layers | 4.97e-3 |
| Fixed Spectral (PCSM w/o Cali) | 5.31e-3 |
| Calibrated Spectral (PCSM) | **4.38e-3** |

To investigate this, we conduct an experiment manually adding high-frequency features to the fixed spectral baseline (i.e. the PCSM w/o Cali) either in compliance or in contradiction with the principle based on spectral gates learned by PCSM. Specifically, we examine the impact of high-frequency boosting in three different layer options. These involve boosting the high frequency in the early layers (1, 2, 3), which differs from PCSM; the late layers (6, 7, 8), somewhat similar to PCSM; and the middle layers (3, 4, 5), identical to PCSM. We implement high-frequency augmentation using Sparse-Frequency Spectral Transform, described in Section A.1.2.

The results, laid out in Table 5, reveal that the model emulating PCSM's exact layer choice (middle layers) exhibits optimal performance. The next best model aligns closely with PCSM's choice (late layers). Conversely, models diverging from PCSM's chosen method (early layers) were inferior in performance. These outcomes suggest that the spectral gates learned by PCSM could guide the frequency design for fixed spectral neural operators.

## 4 RELATED WORK

**Spectral-based Operator Learning.** Stem from Li et al. (2020), numerous works have explored learning operator mappings in spectrum space, which significantly reduces learning difficulty through efficient function approximation with spectral basis function. FNO (Li et al., 2020) learns operators in fourier spectral space, LNO (Cao et al., 2024) learns in laplacian spectral space, and WMT (Gupta et al., 2021) learns in wavelet spectral space. In addition, a line of works investigates the issue of spectral-based neural operators, including the complex physical domain processing (Li et al., 2023c; Bonev et al., 2023; Liu et al., 2024), computational efficiency enhancement (Poli et al., 2022; Tran et al., 2021), multi-scale feature processing (Rahman et al., 2022; Zhang et al., 2024), and

generalization capability improvement (Brandstetter et al., 2022; Yue et al., 2024). Moreover, Li et al. (2024) and Du et al. (2023) explored physics-driven neural operators.

Prior studies, however, employ static spectral feature processing, which restricts the networks' point-level adaptability and makes the network struggle to handle spatially varying phenomena. In contrast, we calibrate the spectral eigenfunctions with the point-wise frequency preference learned by spectral gates, significantly enhancing the feature learning flexibility of spectral-based methods.

**Attention-based Operator Learning.** Recently, learning operator mappings based on attention mechanism (Vaswani, 2017) draws extensive studies. The primary benefits of attention are the capability to handle any physical domains and point-level flexibility for learning high-quality operators for diverse PDEs. To resolve the quadratic complexity of attention mechanism, previous works (Li et al., 2023a; Cao, 2021; Hao et al., 2023) employ efficient attentions for operator learning, and Factformer (Li et al., 2023b) enhance model efficiency with multidimensional factorized attention mechanism. Besides, Liu et al. (2023) introduce a hierarchical transformer based neural operator for learning better multi-scale features, Xiao et al. (2023) alleviate the overfitting of neural operators with orthogonal attention mechanism. In addition, Transolver (Wu et al., 2024) introduces a new operator learning framework by first predicting slices and then learning attentions between different slices.

While attention-based neural operators achieve impressive performance on various PDEs (Wu et al., 2024), their lack of spectral constraints results in subpar performance under limited training data amount and unseen resolutions samples as compared to spectral-based methods. Instead, we develop point-adaptive spectral processing for learning neural operators, thus simultaneously preserving the continuity prior of spectral-based methods and point-level flexibility of attention-based methods.

**Neural Gate Mechanisms**. Gate modules are widely employed in deep neural models. In Mixture-of-Experts models (Jacobs et al., 1991; Shazeer et al., 2017), the gate layer determines which expert networks to activate for processing each input. Besides, gated recurrent unit (Cho, 2014) uses the gate layer to control the flow of information, enabling the model to update the memory cell based on the relevance of the previous and current inputs. Additionally, Dauphin et al. (2017) shows the gate mechanism can help select words or features for next word prediction in language modeling.

This work introduces the concept of Calibrated Spectral Transform via neural gate mechanism, which performs spectral transform integrating point-wise frequency preference. We show its significance for operator learning, by providing point-level flexibility akin to attention mechanism while maintaining the advantages of spectral basis.

## 5 LIMITATION AND FUTURE WORK

Despite obtaining superior performance in a variety of scenarios, the introduced Point-Calibration methods inevitably suffer certain limitations that do not affect the core conclusion of this work, and they are worth further exploration in future works.

- Firstly, the introduced spectral feature calibration technique remains a fundamental design, without considering demands in particular circumstances. Therefore, it is meaningful to develop more customized calibration methods such as learning frequency preference with physical-driven losses.
- Additionally, although this work has explored a broad PDE solving problems, numerous real-world physical systems still warrant further investigation. And it is significant to investigate the application of Point-Calibrated Transform in general deep learning tasks such as time-series signal prediction and computer vision learning.

## 6 CONCLUSION

This work presents Point-Calibrated Spectral Mixer based Neural Operators (PCSM), enabling the point-level adaptivity in spectral-based neural operators by integrating point-wise frequency preference for spectral processing. The proposed Calibrated Spectral Transform holds significant potential applications in numerous spectrum-related deep models. Comprehensive experiments validate the superior performance in various PDE solving scenarios of PCSM, benefiting from the combination of spectral prior of spectral-based methods and point-level adaptivity of attention-based methods. We hope PCSM can provide insights for future exploration of PDE solving tasks.

REPRODUCIBILITY STATEMENT

First, to ensure the reproducibility of performance gains achieved by PCSM, we report the average and standard deviation across three repeated runs in Table. 1 and Table. 2. Additionally, the tool library and benchmarks employed in this work are open-sourced thanks to Li et al. (2020), Li et al. (2023c) and Chen et al. (2023), and we delineate the implementation detail for each problem in Table. 6. Furthermore, to facilitate replication of our experiments, we include the source code for the experiments on Darcy Flow, Navier-Stokes, Airfoil, and Plasticity in supplementary material, and the provided file "README.md" presents the step-by-step running instructions. Extra code will be organized and made available in the next version.

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

# A APPENDIX

## A.1 METHODOLOGY EXTENSION

### A.1.1 FIXED SPECTRAL MIXER

This section presents the implementation of the fixed spectral baseline, i.e. PCSM w/o Cali.

Based on Equation. 5 and the Laplace-Beltrami Transform in Equation. 18 and Equation. 19, the Fixed Spectral Mixer (PCSM w/o Cali) is formulated as follows:

$$\mathcal{F}^{\text{mixer}}_{\text{spectral}}(\boldsymbol{x}) = \mathcal{T}^{-1}_{\text{LBT}} \circ \text{FC} \circ \text{LayerNorm} \circ \mathcal{T}_{\text{LBT}}(\boldsymbol{x}), \tag{13}$$

where $\text{LayerNorm}(\cdot)$ is introduced to normalize the spectral features for more efficient optimization and enhanced generalization. Additionally, we share the learnable weights of FC for all spectrum frequencies. To ensure a fair comparison, we also employ the multi-head design for the fixed spectral baseline. Therefore, the only difference between PCSM (w/o Cali) and PCSM is the spectral calibration operation.

### A.1.2 SPARSE-FREQUENCY FIXED SPECTRAL TRANSFORM

To enhance the learning capability of fixed spectral methods, we attempt to manually add high-frequency spectral features in several network layers. Specifically, instead of using the lowest $N^k$ frequencies, we uniformly take $N^k$ frequencies from the lowest $N^k \times r$ frequencies, where $r$ is the sparsity ratio. Higher $r$ indicates using more high frequencies and we set $r = 2$ and $r = 4$ for different layers.

We find that using Sparse-Frequency Spectral Mixer in partial network layers effectively improves the performance of fixed spectral methods. However, such manual frequency design relies on prior knowledge and repeated experiments to select appropriate layers and sparsity ratios, and additional computational cost is required for calculating LBO eigenfunctions with $N^K \times r$ frequencies. To address this issue, we experiment fixed spectral design guided by learned spectral gates of PCSM, as shown in Table. 5.

### A.1.3 POINT-CALIBRATED FOURIER TRANSFORM

The introduced Point-Calibration technique could also be integrated with additional spectral transform approaches, such as the widely used Fourier Transform in operator learning.

Consider $N$ physical points sampled from the 1-dim Euclidean domain, the latent features could be denoted as $\boldsymbol{x} \in \mathbb{R}^{N \times d_v}$. Discrete Fourier Transform (DFT) could be described as the matrix multiplication between $\boldsymbol{x}$ and $\boldsymbol{F}$, where $\boldsymbol{F} \in \mathbb{R}^{N \times N}$ is defined as $\boldsymbol{F}_{n,k} = e^{-j2\pi nk/N}$. Inverse Discrete Fourier Transform (DFT) is matrix multiplication with $\boldsymbol{F}^*$, where $\boldsymbol{F}^*$ is the conjugate transpose of $\boldsymbol{F}$. Suppose $N^k$ lowest frequencies are employed in the spectral domain, we can write the DFT and IDFT operation as follows:

$$\mathcal{T}_{\text{DFT}}(\boldsymbol{x}) = \boldsymbol{F}^*_{\text{trunk}}\boldsymbol{x} \,, \; \mathcal{T}_{\text{IDFT}}(\boldsymbol{x}) = \boldsymbol{F}_{\text{trunk}}\hat{\boldsymbol{x}}, \tag{14}$$

where $\boldsymbol{F}_{\text{trunk}} \in \mathbb{R}^{N \times N^k}$ and $\boldsymbol{F}^*_{\text{trunk}} \in \mathbb{R}^{N^k \times N}$ are the matrix defined as $\boldsymbol{F}_{\text{trunk}} = \boldsymbol{F}_{[:,0:N^k]}$, $\boldsymbol{F}^*_{\text{trunk}} = \boldsymbol{F}^*_{[0:N^k,:]}$. $\hat{\boldsymbol{x}} \in \mathbb{R}^{N^k \times d_v}$ is the truncated spectrum representation of $\boldsymbol{x}$.

Next, with the point-wise frequency preference $\boldsymbol{G}$ shown in Equation. 8, we can obtain the Point-Calibrated Fourier Transform as follows:

$$\mathcal{T}_{\text{PC-DFT}}(\boldsymbol{x}) = (\boldsymbol{G} \odot \boldsymbol{F}^*_{\text{trunk}})\boldsymbol{x} \,, \; \mathcal{T}_{\text{PC-IDFT}}(\boldsymbol{x}) = (\boldsymbol{G} \odot \boldsymbol{F}_{\text{trunk}})\hat{\boldsymbol{x}}. \tag{15}$$

$\mathcal{T}_{\text{PC-DFT}}(\cdot)$ and $\mathcal{T}_{\text{PC-IDFT}}(\cdot)$ could be seamlessly integrated with spectral mixer for constructing Point-Calibrated Fourier Neural Operators.

## A.2 DEFINITIONS AND THEORETICAL FOUNDATIONS

This section provides the definitions and theoretical foundations of the introduced Point-Calibrated Spectral Mixer. We first present the formal definition of Point-Calibrated Spectral Transform and

its matrix multiplication form in Section. A.2.1. Then we introduce the preliminary lemmas about the neural operator learning in Section. A.2.2. Next, we provide the theoretical demonstration (Theory. A.8) that PCSM is the learnable integral neural operator (Theorem. A.8) in Section. A.2.3. We demonstrate that PCSM integrates the kernel functions of previous linear attention methods and spectral methods (Remark. A.9).

### A.2.1 DEFINITION OF POINT-CALIBRATED SPECTRAL TRANSFORM

**Definition A.1.** *Laplace-Beltrami Spectral Transform.*

Consider the input function $\boldsymbol{u} : \Omega \to \mathbb{R}^d$ defined on the physical domain $\Omega$, where $\Omega \subset \mathbb{R}^{d_g}$ and $d_g$ represents the dimension of physical space. For numerical calculation, we commonly take discrete points from $\Omega$. We denote the set of sampled points from $\Omega$ as $\Omega'$, i.e. $\Omega' \subset \Omega$. The number of sampled points in $\Omega'$ is denoted as $N$. In addition, we denote the spectral transform of the function $\boldsymbol{u}$ as $\hat{\boldsymbol{u}} : D \to \mathbb{R}^d$. Here $D$ represents the spectral space and $D \subset \mathbb{R}$. We take discrete frequencies from $D$ and note the set of sampled frequencies as $D'$. The number of elements in $D'$ is denoted with $N^k$. With these notations, we formulate the Laplace-Beltrami Spectral Transform and inverse Laplace-Beltrami Spectral Transform as follows:

$$\mathcal{T}_{\text{LBT}}(\boldsymbol{u})(k) = \sum_{x \in \Omega'} \phi_k(x) \cdot \boldsymbol{u}(x), \tag{16}$$

$$\mathcal{T}_{\text{LBT}}^{-1}(\hat{\boldsymbol{u}})(x) = \sum_{k \in D'} \phi_k(x) \cdot \hat{\boldsymbol{u}}(k), \tag{17}$$

where $x \in \Omega'$ and $k \in D'$ are the elements in spatial space $\Omega'$ and spectral space $D'$ respectively. $\phi_k(\cdot)$ is the LBO eigenfunction of frequency $k$. The LBO eigenfunctions $\boldsymbol{\phi} = [\phi_1, \phi_2, ..., \phi_{N^k}]$ are calculated based on the sampled points in physical domain $\Omega'$. Specifically, we calculate the eigenfunctions using the open-sourced package robust-laplacian (Sharp & Crane, 2020) library [2].

For convenient model implementation, we can also write the spectral transform in matrix multiplication form. Specifically, taking the discrete points in $\Omega'$ and discrete frequencies in $D'$, the functions could be represented as matrix, specifically $\boldsymbol{u} \in \mathbb{R}^{N \times d}$, $\hat{\boldsymbol{u}} \in \mathbb{R}^{N^k \times d}$, $\phi_k \in \mathbb{R}^{N \times 1}$ and $\boldsymbol{\phi} \in \mathbb{R}^{N \times N^k}$.

$$\mathcal{T}_{\text{LBT}}(\boldsymbol{u}) = [\boldsymbol{u}^T \phi_1, \boldsymbol{u}^T \phi_2, ..., \boldsymbol{u}^T \phi_{N^k}]^T = \boldsymbol{\phi}^T \boldsymbol{u}, \tag{18}$$

$$\mathcal{T}_{\text{LBT}}^{-1}(\hat{\boldsymbol{u}}) = [\phi_1, \phi_2, ..., \phi_{N^k}]\hat{\boldsymbol{u}} = \boldsymbol{\phi}\hat{\boldsymbol{u}}. \tag{19}$$

This is consistent with Figure. 1, where the spectral transform and inverse spectral transform are implemented with simple matrix multiplications.

**Definition A.2.** *Point-Calibrated Laplace-Beltrami Spectral Transform.*

Following Definition. A.1, we formulate the Point-Calibrated Laplace-Beltrami Spectral Transform as follows:

$$\mathcal{T}_{\text{PC-LBT}}(\boldsymbol{u})(k) = \sum_{x \in \Omega'} \phi_k(x) \cdot \boldsymbol{g}_k(x) \cdot \boldsymbol{u}(x), \tag{20}$$

$$\mathcal{T}_{\text{PC-LBT}}^{-1}(\hat{\boldsymbol{u}})(x) = \sum_{k \in D'} \phi_k(x) \cdot \boldsymbol{g}_k(x) \cdot \hat{\boldsymbol{u}}(k). \tag{21}$$

Here $\boldsymbol{g}_k(x) = \boldsymbol{G}_k(u)(x)$ is the learnable gate value. We implement $\boldsymbol{G}(u) \in \mathbb{R}^{|\Omega'| \times |D'|}$ with an element-wise MLP layer and the Softmax activation function as shown in Equation. 8. $\boldsymbol{G}_k$ is the $k$-th column of $\boldsymbol{G}$.

Similar to Definition 1, we write the spectral transform in matrix multiplication form for convenient implementation. Specifically, we take the matrix form of these functions, $\boldsymbol{u} \in \mathbb{R}^{N \times d}$, $\hat{\boldsymbol{u}} \in \mathbb{R}^{N^k \times d}$, $\phi_k \in \mathbb{R}^{N \times 1}$, $\boldsymbol{\phi} \in \mathbb{R}^{N \times N^k}$, $\boldsymbol{g}_k \in \mathbb{R}^{N \times 1}$ and $\boldsymbol{G} \in \mathbb{R}^{N \times N^k}$.

$$\mathcal{T}_{\text{PC-LBT}}(\boldsymbol{u}) = [\boldsymbol{u}^T(\boldsymbol{g}_1 \odot \phi_1), \boldsymbol{u}^T(\boldsymbol{g}_2 \odot \phi_2), ..., \boldsymbol{u}^T(\boldsymbol{g}_{N^k} \odot \phi_{N^k})]^T = (\boldsymbol{\phi} \odot \boldsymbol{G})^T \boldsymbol{u}, \tag{22}$$

$$\mathcal{T}_{\text{PC-LBT}}^{-1}(\hat{\boldsymbol{u}}) = [\boldsymbol{g}_1 \odot \phi_1, \boldsymbol{g}_2 \odot \phi_2, ..., \boldsymbol{g}_{N^k} \odot \phi_{N^k}]\hat{\boldsymbol{u}} = (\boldsymbol{\phi} \odot \boldsymbol{G})\hat{\boldsymbol{u}}. \tag{23}$$

---

[2]Robust-laplacian library link: https://github.com/nmwsharp/nonmanifold-laplacian

This is also consistent with Figure. 1, where the spectral transform and inverse spectral transform are implemented with simple matrix multiplications.

### A.2.2 PRELIMINARY THEOREM: INTEGRAL NEURAL OPERATOR LEARNING

The following theorems are summarized from previous works (Li et al., 2020; Kovachki et al., 2023; Wu et al., 2024), which provide the theoretical basis of the proposed Point-Calibrated Spectral Mixer.

**Remark A.3.** *PDEs could be solved by learning integral neural operators.*

Kovachki et al. (2023) formulate the common architecture of neural operators for PDE solving as a stack of network layers.

$$\mathcal{G}_\theta = Q \circ \sigma(W_l + \mathcal{K}_l) \circ \cdots \circ \sigma(W_i + \mathcal{K}_i) \circ \cdots \circ \sigma(W_1 + \mathcal{K}_1) \circ P, \tag{24}$$

where $P$ and $Q$ are both linear point-wise projectors as shown in Equation. 2. $W_i$ is the point-wise fully connected layer and $\mathcal{K}_i$ is the non-local integral operator.

In each network layer, the key is to learn the non-local integral operator $\mathcal{K}_i$ defined as follows:

$$\mathcal{K}_i(\boldsymbol{u})(x) = \int_\Omega \kappa_i(x, \xi, \boldsymbol{u}(x), \boldsymbol{u}(\xi))\boldsymbol{u}(\xi)d\xi, \tag{25}$$

where $\boldsymbol{u}$ is the input function and $\Omega$ is the physical domain, as defined in Section. A.2.1. As presented in (Kovachki et al., 2023), the learnable integral kernel operator enables the mapping between continuous functions, similar to the weight matrix operation that enables the mapping between discrete vectors. It could be demonstrated that various neural operators (Li et al., 2020; Cao, 2021; Chen et al., 2023; Wu et al., 2024) are learning different kernel functions of the stacked integral neural operators shown in Equation. 25.

**Lemma A.4.** *FNO (Li et al., 2020) learns integral neural operators.*

This is demonstrated in Li et al. (2020) and Kovachki et al. (2023). By setting the kernel function as $\kappa(x, \xi, \boldsymbol{u}(x), \boldsymbol{u}(\xi)) = \kappa(x - \xi)$, it could be demonstrated that the kernel integral operator could be implemented with Fourier Transform. For more details you can refer to Li et al. (2020).

**Lemma A.5.** *The standard Transformer (Vaswani, 2017) learns integral neural operators.*

Kovachki et al. (2023) demonstrates that the canonical attention mechanism (Vaswani, 2017) is a special case of integral neural operators. This could be demonstrated by setting the kernel function as follows:

$$\kappa(x, \xi, \boldsymbol{u}(x), \boldsymbol{u}(\xi)) = (\int_\Omega \exp(\boldsymbol{W}_q \boldsymbol{u}(\xi^{'})(\boldsymbol{W}_k \boldsymbol{u}(x))^T)d\xi^{'})^{-1}\exp(\boldsymbol{W}_q \boldsymbol{u}(x)(\boldsymbol{W}_k \boldsymbol{u}(\xi))^T)\boldsymbol{R}, \tag{26}$$

where $\boldsymbol{W}_q \in \mathbb{R}^{d \times d}$, $\boldsymbol{W}_k \in \mathbb{R}^{d \times d}$ and $\boldsymbol{R} \in \mathbb{R}^{d \times d}$ are all the training parameter of the neural network. For simplification, we eliminate the division operation with $\sqrt{d}$. With this formulation, we can derive the attention mechanism based on the kernel integral operator shown in Equation. 25 and Monte-Carlo approximation. The proof can be found in Kovachki et al. (2023). Therefore, the attention mechanism could be employed for PDE solving.

### A.2.3 POINT-CALIBRATED SPECTRAL MIXER AS INTEGRAL NEURAL OPERATORS

To validate that the proposed Point-Calibrated Spectral Mixer (PCSM) could learn the neural operators for PDE solving, this section provides the theoretical demonstration that PCSM is a learnable integral neural operator (Theorem. A.8).

In addition, we notice that PCSM intrinsically integrates attention models and spectral models (Remark. A.9) from the perspective of integral kernels. To illustrate this, we will begin from the demonstrations that the linear attention (Katharopoulos et al., 2020) and the fixed spectral method NORM (Chen et al., 2023) are also integral operators.

**Lemma A.6.** *Linear attention (Katharopoulos et al., 2020) is a learnable integral operator.*

*Proof.* First, we introduce the formulation of the linear attention mechanism shown in Katharopoulos et al. (2020). Given the input sequential features $\boldsymbol{x} \in \mathbb{R}^{N \times d}$, where $N$ and $d$ represent the length of

the sequence and the dimension of feature embeddings, respectively. We consider the linear attention mechanism from Katharopoulos et al. (2020) with following formulation:

$$\text{Linear-Attention}(\boldsymbol{x}) = \psi(\boldsymbol{W}_q \boldsymbol{x})\,(\psi(\boldsymbol{W}_k \boldsymbol{x})^T\,\boldsymbol{W}_v \boldsymbol{x}), \tag{27}$$

where $\boldsymbol{W}_q \in \mathbb{R}^{d \times d}$, $\boldsymbol{W}_k \in \mathbb{R}^{d \times d}$ and $\boldsymbol{W}_v \in \mathbb{R}^{d \times d}$ are all the training parameters of the neural network, similar to the standard attention mechanism. $\psi(\cdot)$ is the normalization function applied to each row of matrix $\boldsymbol{W}_q \boldsymbol{x}$ and $\boldsymbol{W}_k \boldsymbol{x}$. Katharopoulos et al. (2020) instantiates $\psi(\cdot) = \text{elu}(x) + 1$, where $\text{elu}(\cdot)$ is the exponential linear unit (Clevert, 2015). As shown in Equation. 27, the core factor of linear attention is that we first calculate the matrix multiplication between $(\psi(\boldsymbol{W}_k \boldsymbol{x})^T$ and $\boldsymbol{W}_v \boldsymbol{x}$, eliminating the expansion of full attention matrix.

Next, we will demonstrate the linear attention mechanism is an integral neural operator. Consider the input function $\boldsymbol{u} : \Omega \rightarrow \mathbb{R}^d$ as Section. A.2.1. Let's set the kernel function as follows:

$$\kappa(x, \xi, \boldsymbol{u}(x), \boldsymbol{u}(\xi)) = \rho(\boldsymbol{u}(x), \boldsymbol{u}(\xi))\,\boldsymbol{R}, \tag{28}$$

$$\rho(\boldsymbol{u}(x), \boldsymbol{u}(\xi)) = <\psi(\boldsymbol{W}_q \boldsymbol{u}(x)), \psi(\boldsymbol{W}_k \boldsymbol{u}(\xi)) >, \tag{29}$$

where $\boldsymbol{W}_q \in \mathbb{R}^{d \times d}$ and $\boldsymbol{W}_v \in \mathbb{R}^{d \times d}$ are both free parameters, and $\psi(\cdot)$ is the normalization function. $\boldsymbol{R} \in \mathbb{R}^{d \times d}$ is also the matrix of parameters.

Based on the integral neural operator shown in Equation. 25, we could derive the linear attention mechanism shown in Equation. 27:

$$
\begin{aligned}
\mathcal{K}(\boldsymbol{u})(x) &= \int_\Omega \kappa(x, \xi, \boldsymbol{u}(x), \boldsymbol{u}(\xi))\boldsymbol{u}(\xi)d\xi \\
&= \int_\Omega < \psi(\boldsymbol{W}_q \boldsymbol{u}(x)), \psi(\boldsymbol{W}_k \boldsymbol{u}(\xi)) > \boldsymbol{R}\boldsymbol{u}(\xi)d\xi \quad \text{(Equation. 28)} \\
&= \int_\Omega \psi(\boldsymbol{W}_q \boldsymbol{u}(x))\psi(\boldsymbol{W}_k \boldsymbol{u}(\xi))^T \boldsymbol{W}_v \boldsymbol{u}(\xi)d\xi \quad \text{(Matrix } \boldsymbol{R} \text{ as matrix } \boldsymbol{W}_v) \\
&\approx \sum_{\xi \in \Omega'} \psi(\boldsymbol{W}_q \boldsymbol{u}(x))\psi(\boldsymbol{W}_k \boldsymbol{u}(\xi))^T \boldsymbol{W}_v \boldsymbol{u}(\xi) \quad \text{(Monte-Carlo approximation)} \\
&= \psi(\boldsymbol{W}_q \boldsymbol{u}(x)) \sum_{\xi \in \Omega'} \psi(\boldsymbol{W}_k \boldsymbol{u}(\xi))^T \boldsymbol{W}_v \boldsymbol{u}(\xi)
\end{aligned}
\tag{30}
$$

where $\Omega'$ is the set of sampled points from $\Omega$, as presented in Definition. A.1. Same as Lemma. A.5, we utilize the Monte-Carlo approximation used in Kovachki et al. (2023), which is based on sufficient sampling points from the physical domain $\Omega$ and proper discretization of the normalization function.

By taking the discrete matrix form of the function $\boldsymbol{u}$ like Definition. A.1, we can get the equation form that is the same as the implementation of linear attention (Equation. 27). Therefore, linear attention could learn the integral neural operators for PDE solving.

**Lemma A.7.** *The spectral model NORM (Chen et al., 2023) learns integral neural operators.*

*Proof.* First, we present the spectral mixer used in NORM. NORM (Chen et al., 2023) proposes to learn neural operators through Laplace-Beltrami Spectral Transform. The network consists of stacked layers based on the spectral mixer as follows:

$$\mathcal{F}_{\text{spectral}}^{\text{mixer}}(\boldsymbol{u}) = \mathcal{T}_{\text{LBT}}^{-1} \circ \text{Project} \circ \mathcal{T}_{\text{LBT}}(\boldsymbol{u}), \tag{31}$$

where $\boldsymbol{u} : \Omega \rightarrow \mathbb{R}^d$ is the input function, $\mathcal{T}_{\text{LBT}}$ and $\mathcal{T}_{\text{LBT}}^{-1}$ represent the Laplace-Beltrami Spectral Transform and inverse Laplace-Beltrami Spectral Transform presented in Definition. A.1. Project represents the processing in spectral space, and it could be implemented in multiple methods. In NORM (Chen et al., 2023) it is a fully connected layer like FNO (Li et al., 2020).

Next, we could set the kernel function shown in Equation. 25 as follows:

$$\kappa(x, \xi, \boldsymbol{u}(x), \boldsymbol{u}(\xi)) = \rho(x, \xi)\,\boldsymbol{R}, \tag{32}$$

$$\rho(x, \xi) = \int_D \phi_k(x) \cdot \phi_k(\xi)dk, \tag{33}$$

where $\phi_k(x)$ is the spectral eigenfunction of the physical domain $\Omega$, as presented in Definition. A.1. $D$ represents the spectral space. $\boldsymbol{R} \in \mathbb{R}^{d \times d}$ is the matrix of parameters.

Based on the kernel function, we could derive the spectral mixer based on the Laplace-Beltrami Transform used in NORM (Chen et al., 2023) from the integral neural operator shown in Equation. 25:

$$
\begin{aligned}
\mathcal{K}(\boldsymbol{u})(x) &= \int_{\Omega} \kappa(x, \xi, \boldsymbol{u}(x), \boldsymbol{u}(\xi)) \boldsymbol{u}(\xi) d\xi \\
&= \int_{\Omega} \int_{D} \phi_k(x) \cdot \phi_k(\xi) dk \, \boldsymbol{R} \boldsymbol{u}(\xi) d\xi && \text{(Equation. 32)} \\
&= \int_{D} \phi_k(x) \boldsymbol{R} \int_{\Omega} \phi_k(\xi) \boldsymbol{u}(\xi) d\xi dk \\
&\approx \sum_{k \in D'} \left( \phi_k(x) \cdot \boldsymbol{R} \sum_{\xi \in \Omega'} \phi_k(\xi) \cdot \boldsymbol{u}(\xi) \right) && \text{(Monte-Carlo approximation)} \\
&= \sum_{k \in D'} \left( \phi_k(x) \cdot \boldsymbol{R} \, \mathcal{T}_{\text{LBT}}(\boldsymbol{u})(k) \right) && \text{(Equation. 16)} \\
&= \mathcal{T}_{\text{LBT}}^{-1}(\boldsymbol{R} \, \mathcal{T}_{\text{LBT}}(\boldsymbol{u}))(x) && \text{(Equation. 17)} \\
&= (\mathcal{T}_{\text{LBT}}^{-1} \circ \text{Project} \circ \mathcal{T}_{\text{LBT}}(\boldsymbol{u}))(x), && \text{(Matrix multiplication as Project)}
\end{aligned}
\tag{34}
$$

where $\Omega'$ is the set of sampled points from $\Omega$, as presented in Definition. A.1. Same as Lemma. A.5, we use the Monte-Carlo approximation used in Kovachki et al. (2023), which is based on sufficient sampling points from the physical domain $\Omega$ and proper discretization of the spectral function $\phi_k(\cdot)$.

The final form is exactly same as the spectral mixer used in NORM (Chen et al., 2023) as shown in Equation. 31. Therefore, NORM (Chen et al., 2023) could learn the integral neural operators for PDE solving.

**Theorem A.8.** *Point-Calibrated Spectral Mixer (PCSM) is the integral neural operator.*

*Proof.* The Point-Calibrated Spectral Mixer is represented in the following form:

$$
\mathcal{F}_{\text{spectral}}^{\text{mixer}}(\boldsymbol{u}) = \mathcal{T}_{\text{PC-LBT}}^{-1} \circ \text{Project} \circ \mathcal{T}_{\text{PC-LBT}}(\boldsymbol{u}),
\tag{35}
$$

where $\boldsymbol{u} : \Omega \rightarrow \mathbb{R}^d$ is the input function, $\mathcal{T}_{\text{PC-LBT}}$ and $\mathcal{T}_{\text{PC-LBT}}^{-1}$ represent the Point-Calibrated Laplace-Beltrami Spectral Transform and inverse Point-Calibrated Laplace-Beltrami Spectral Transform respectively, formulated in Definition. A.2. Project represents the processing in spectral space, and it could be implemented in multiple methods. In PCSM, it consists of a point-wise normalization and a fully connected layer.

We could set the kernel function shown in Equation. 25 as follows:

$$
\kappa(x, \xi, \boldsymbol{u}(x), \boldsymbol{u}(\xi)) = \rho(x, \xi, \boldsymbol{u}(x), \boldsymbol{u}(\xi)) \, \boldsymbol{R},
\tag{36}
$$

$$
\rho(x, \xi, \boldsymbol{u}(x), \boldsymbol{u}(\xi)) = \int_{D} \phi_k(x) \cdot \phi_k(\xi) \cdot \boldsymbol{g}_k(x) \cdot \boldsymbol{g}_k(\xi) dk,
\tag{37}
$$

where $\boldsymbol{g}_k(\cdot) \in \mathbb{R}$ is the gate value introduced in Definition. A.2. $\boldsymbol{\phi}(x) = [\phi_1(x), \phi_2(x), ..., \phi_{|D|}(x)]$ is the set of eigenfunctions computed on the physical domain $\Omega$, as presented in Definition. A.1, and $D$ represents the spectral space. $\boldsymbol{R} \in \mathbb{R}^{d \times d}$ is a parameterized matrix.

Next, we can arrive Point-Calibrated Spectral Mixer from the kernel function in the above equation and the formulation of the integral neural operator (Equation. 25). The procedure is shown as follows:

$$
\begin{aligned}
\mathcal{K}(\boldsymbol{u})(x) &= \int_{\Omega} \kappa(x, \xi, \boldsymbol{u}(x), \boldsymbol{u}(\xi)) \boldsymbol{u}(\xi) d\xi \\
&= \int_{\Omega} \int_{D} \phi_k(x) \cdot \phi_k(\xi) \cdot \boldsymbol{g}_k(x) \cdot \boldsymbol{g}_k(\xi) dk \, \boldsymbol{R} \boldsymbol{u}(\xi) d\xi \quad \text{(Equation. 36)} \\
&= \int_{D} \phi_k(x) \boldsymbol{g}_k(x) \boldsymbol{R} \int_{\Omega} \phi_k(\xi) \boldsymbol{g}_k(\xi) \boldsymbol{u}(\xi) d\xi dk \\
&\approx \sum_{k \in D'} \left( \phi_k(x) \boldsymbol{g}_k(x) \cdot \boldsymbol{R} \sum_{\xi \in \Omega'} \phi_k(\xi) \boldsymbol{g}_k(\xi) \cdot \boldsymbol{u}(\xi) \right) \quad \text{(Monte-Carlo approximation)} \\
&= \sum_{k \in D'} \left( \phi_k(x) \cdot \boldsymbol{R} \, \mathcal{T}_{\text{PC-LBT}}(\boldsymbol{u})(k) \right) \quad \text{(Equation. 20)} \\
&= \mathcal{T}_{\text{PC-LBT}}^{-1}(\boldsymbol{R} \, \mathcal{T}_{\text{PC-LBT}}(\boldsymbol{u}))(x) \quad \text{(Equation. 21)} \\
&= (\mathcal{T}_{\text{PC-LBT}}^{-1} \circ \text{Project} \circ \mathcal{T}_{\text{PC-LBT}}(\boldsymbol{u}))(x), \quad \text{(Matrix multiplication as Project)}
\end{aligned}
\tag{38}
$$

where $\Omega'$ is the set of sampled points from $\Omega$, as presented in Definition. A.1. Same as Lemma. A.5, we use the Monte-Carlo approximation used in Kovachki et al. (2023), which is based on sufficient sampling points from the physical domain $\Omega$ and proper discretization of the spectral function $\phi_k(\cdot)$. We note that although the Project is a LayerNorm and fully connected layer, the matrix multiplication could act as the Project. This is because the point-wise normalization in spectral space could also be represented as the multiplication with a matrix that is irrelevant to spatial values.

Therefore, the introduced Point-Calibrated Spectral Neural Operator is essentially an integral neural operator presented in Kovachki et al. (2023).

**Remark A.9.** *PCSM integrates the kernels of attention-based models and spectral-based models. If we remove the component of spectral methods from the kernel function, PCSM will become the linear attention mechanism. If we remove the component of attention methods from the kernel function, PCSM will become the classic spectral mixer.*

*Proof.* The kernel function (Equation. 36) employed in PCSM actually combines kernel functions of linear attention (Katharopoulos et al., 2020) and NORM (Chen et al., 2023), as shown in the equation below:

$$
\begin{aligned}
\rho(x, \xi, \boldsymbol{u}(x), \boldsymbol{u}(\xi)) &= \int_{D} \phi_k(x) \cdot \phi_k(\xi) \cdot \boldsymbol{g}_k(x) \cdot \boldsymbol{g}_k(\xi) dk, \\
&= \int_{D} \underbrace{\phi_k(x) \cdot \phi_k(\xi)}_{\text{Spectral Component}} \cdot \underbrace{\psi(\boldsymbol{W}_g \boldsymbol{u}(x))(k) \cdot \psi(\boldsymbol{W}_g \boldsymbol{u}(\xi))(k)}_{\text{Attention Component}} dk,
\end{aligned}
\tag{39}
$$

where $W_g$ represents the parameterized matrix for the point-wise spectral gates prediction, and $\psi(\cdot)$ is the Softmax function. They are the spectral gates module used in PCSM.

If we remove the spectral component, we will get the kernel function used in the linear attention mechanism:

$$
\begin{aligned}
\rho(x, \xi, \boldsymbol{u}(x), \boldsymbol{u}(\xi)) &= \int_{D} \underbrace{\cancel{\phi_k(x) \cdot \phi_k(\xi)}}_{\text{Spectral Component}} \cdot \underbrace{\psi(\boldsymbol{W}_g \boldsymbol{u}(x))(k) \cdot \psi(\boldsymbol{W}_g \boldsymbol{u}(\xi))(k)}_{\text{Attention Component}} dk \\
&= \int_{D} \psi(\boldsymbol{W}_g \boldsymbol{u}(x))(k) \cdot \psi(\boldsymbol{W}_g \boldsymbol{u}(\xi))(k) dk \\
&= \underbrace{< \psi(\boldsymbol{W}_g \boldsymbol{u}(x)), (\psi(\boldsymbol{W}_g \boldsymbol{u}(\xi))) >}_{\text{Same as Equation. 29}}.
\end{aligned}
\tag{40}
$$

From this kernel function, we can derive the linear attention mechanism, as shown in Equation. 30. The difference to Katharopoulos et al. (2020) is the varied instantiations of $\boldsymbol{W}_q$, $\boldsymbol{W}_v$ (the weights are shared and the output dimension is set as $N^K$), and the activation function $\psi$ (we use Softmax function).

If we remove the attention component, we will get the kernel function used in the fixed spectral method NORM (Chen et al., 2023):

$$
\begin{aligned}
\rho(x, \xi, \boldsymbol{u}(x), \boldsymbol{u}(\xi)) &= \int_D \underbrace{\phi_k(x) \cdot \phi_k(\xi)}_{\text{Spectral Component}} \cdot \underbrace{\psi(\boldsymbol{W}_g \boldsymbol{u}(x))(k) \cdot \psi(\boldsymbol{W}_g \boldsymbol{u}(\xi))(k)}_{\text{Attention Component}} \, dk \\
&= \underbrace{\int_D \phi_k(x) \cdot \phi_k(\xi) dk}_{\text{Same as Equation. 33}}
\end{aligned}
\tag{41}
$$

From this kernel function, we can derive the spectral method NORM (Chen et al., 2023), as shown in Equation. 34.

### A.3 EXPERIMENT SETUPS

#### A.3.1 IMPLEMENTATION DETAIL

We implement PCSM with close parameter amount with the compared baselines (Hao et al., 2023; Wu et al., 2024; Chen et al., 2023). The same optimizer setup with Transolver (Wu et al., 2024) is employed. All experiments could be conducted with a single A100 device. The implementation detail for each problem is presented in Table. 6.

#### A.3.2 METRIC

Same as previous works (Li et al., 2020; Wu et al., 2024), the assessed metric in this work is the Relative L2 Error, formulated as follows:

$$
L2 = \frac{1}{N_{\text{test}}} \sum_{i=1}^{N_{\text{test}}} \frac{\|\hat{u}_i - u_i\|_2}{\|u_i\|_2},
\tag{42}
$$

where $N_{\text{test}}$ is the number of evaluated samples, $\hat{u}_i$ represents the predicted trajectory, and $u_i$ denotes the ground-truth trajectory.

#### A.3.3 EVALUATED PDE PROBLEMS

**Darcy Flow.** Darcy Flow is a steady-state solving problem from Li et al. (2020). We experiment with the identical setup as previous works (Li et al., 2020; Tran et al., 2021; Wu et al., 2024). The resolution of input and output functions are $85 \times 85$ and there are 1000 trajectories for training and an additional 200 data for testing.

**Navier-Stokes.** Navier-Stokes is the PDE solving problem introduced in FNO (Li et al., 2020). We experiment with the most challenging split where the viscosity coefficient is 1e-5. The input is the vorticity field of the first 10 time steps and the target is to predict the status of the following 10 steps. The training and test amounts are 1000 and 200 respectively.

**Airfoil.** Airfoil is an irregular domain problem from Geo-FNO (Li et al., 2023c). In this experiment, the neural operators take the airfoil shape as input and predict the Mach number on the domain. The irregular domain is represented as structured meshes aligned with standard rectangles. All airfoil shapes come from the NACA-0012 case by the National Advisory Committee for Aeronautics. 1000 samples are used for training and additional 200 samples are used for evaluation.

**Plasticity.** This task requires neural operators to predict the deformation state of plasticity material and the impact from the upper boundary by an irregular-shaped rigid die. The input is the shape of the die and the output is the deformation of each physical point in four directions in future 20 time steps. There are 900 data for training and an additional 80 data for testing.

**Irregular Darcy.** This problem involves solving the Darcy Flow equation within an irregular domain. The function input is $a(x)$, representing the diffusion coefficient field, and the output $u(x)$ represents the pressure field. The domain is represented by a triangular mesh with 2290 nodes. The neural operators are trained on 1000 trajectories and tested on an extra 200 trajectories.

**Pipe Turbulence.** Pipe Turbulence system is modeled by the Navier-Stokes equation, with an irregular pipe-shaped computational domain represented as 2673 triangular mesh nodes. This task

Table 6: Implementation detail for each PDE problem.

| Problems | Model Configurations | | | | Training Configurations | | | | | |
|---|---|---|---|---|---|---|---|---|---|---|
| | Depth | Width | Head Number | $N^k$ | Optimizer | Scheduler | Initial Lr | Weight Decay | Epochs | Batch Size |
| Darcy Flow | 8 | 128 | 8 | 128 | AdamW | OneCycleLR | 1e-3 | 1e-5 | 500 | 4 |
| Airfoil | 8 | 128 | 8 | 128 | AdamW | OneCycleLR | 1e-3 | 1e-5 | 500 | 4 |
| Navier-Stokes | 8 | 256 | 8 | 128 | AdamW | OneCycleLR | 1e-3 | 1e-5 | 500 | 4 |
| Plasticity | 8 | 128 | 8 | 128 | AdamW | OneCycleLR | 1e-3 | 1e-5 | 500 | 8 |
| Irregular Darcy | 4 | 64 | 4 | 64 | AdamW | OneCycleLR | 1e-3 | 1e-5 | 2000 | 16 |
| Pipe Turbulence | 4 | 64 | 4 | 64 | AdamW | OneCycleLR | 1e-3 | 1e-5 | 2000 | 16 |
| Heat Transfer | 4 | 64 | 4 | 64 | AdamW | OneCycleLR | 1e-3 | 1e-5 | 2000 | 16 |
| Composite | 4 | 64 | 4 | 64 | AdamW | OneCycleLR | 1e-3 | 1e-5 | 2000 | 16 |
| Blood Flow | 4 | 64 | 4 | 32 | AdamW | OneCycleLR | 1e-3 | 1e-5 | 2000 | 4 |

Table 7: Ablation on prediction module for spectral gates.

| Input Condition | | | Activation Function | | Calibration Level | | Relative Error |
|---|---|---|---|---|---|---|---|
| $x$ | *spectral coefficient* | *x+spectral coefficient* | *sigmoid* | *softmax* | *points* | *global* | |
| ✓ | | | | ✓ | ✓ | | **4.38e-3** |
| | ✓ | | | ✓ | ✓ | | 5.96e-3 |
| | | ✓ | | ✓ | ✓ | | 4.65e-3 |
| ✓ | | | ✓ | | ✓ | | 4.86e-3 |
| ✓ | | | | ✓ | | ✓ | 5.47e-3 |

requires the neural operator to predict the next frame's velocity field based on the previous one. Same as Chen et al. (2023), we utilize 300 trajectories for training and then test the models on 100 samples.

**Heat Transfer.** This problem is about heat transfer events triggered by temperature variances at the boundary. Guided by the Heat equation, the system evolves over time. The neural operator strives to predict 3-dimensional temperature fields after 3 seconds given the initial boundary temperature status. The output domain is represented by triangulated meshes of 7199 nodes. The neural operators are trained on 100 data sets and evaluated on another 100 data.

**Composite.** This problem involves predicting deformation fields under high-temperature stimulation, a crucial factor in composite manufacturing. The trained operator is anticipated to forecast the deformation field based on the input temperature field. The structure studied in this paper is an air-intake component of a jet composed of 8232 nodes, as referenced in (Chen et al., 2023). The training involved 400 data, and the test examined 100 data.

**Blood Flow.** The objective is to foresee blood flow within the aorta, including 1 inlet and 5 outlets. The flow of blood is deemed a homogeneous Newtonian fluid. The computational domain, entirely irregular, is visualized by 1656 triangle mesh nodes. Over a simulated 1.21-second duration, with 0.01-second temporal steps, the neural operator predicts different times' velocity fields given velocity boundaries at the inlet and pressure boundaries at the outlet. Same as (Chen et al., 2023), our experiment involves training on 400 data sets and testing on 100 data.

## A.4 ABLATION STUDY

This section ablates the core modules of PCSM, to reveal the main factors affecting PCSM performance.

**Frequency Preference Prediction Module Ablation.** In Table. 7, we attempt different module designs for frequency preference prediction on Darcy Flow problem.

First, we compare different input conditions for gates prediction, including only $x$, only *spectral coefficient*, and *x+spectral coefficient*. The results are shown in the first 3 lines of Table. 7. Compared to the other two, using only *spectral coefficient* performs poorly. This validates the conclusion that frequency preference learning based on point status is particularly

Table 8: Performance of different frequency numbers on Darcy Flow.

| Problem | Frequency Number | PCSM (w/o Cali) | PCSM (w/ Cali) |
|---|---|---|---|
| Darcy Flow | 16 | 1.04e-2 | **5.10e-3** |
| | 32 | 8.08e-3 | **5.02e-3** |
| | 64 | 6.15e-3 | **4.85e-3** |
| | 128 | 5.31e-3 | **4.38e-3** |
| Navier-Stokes | 16 | 1.18e-1 | **9.51e-2** |
| | 32 | 1.05e-1 | **8.77e-2** |
| | 64 | 9.47e-2 | **7.44e-2** |
| | 128 | 8.37e-2 | **7.28e-2** |

important, which further explains why PCSM could greatly outperform previous fixed spectral-based methods that select frequencies without considering physical status.

Next, we experiment with different activation functions in the gate prediction module, including sigmoid(·) and softmax(·). The results are shown in the first and fourth lines of Table. 7. It could be observed that softmax(·) is better than sigmoid(·). We posit this is due to the normalization operation in softmax(·) ensuring consistent spectral feature scale across different physical points.

In addition, we compare the performance between different gate levels, i.e. "point-level frequency calibration" or "global-level frequency selection", The results are presented in the first and the fifth lines of Table. 7. "Global" refers to first conducting average pooling on physical points and then predicting shared spectral gates for all points. The performance of the "Global" model drops a lot. This directly demonstrates the importance of point-level adaptive feature learning.

**Performance without Gates.** To fully validate the influence of calibration, for all experiments, we compare the performance between the fixed spectral baseline (i.e. "PCSM w/o Cali") and Point-Calibrated Spectral Mixer. The results are presented in Table. 1, 2, 3, 4. Across all problems and setups, PCSM consistently outperforms the fixed spectral counterpart, which validates the significance of spectral calibration for neural operator learning on various PDE solving scenarios.

**Frequency Number.** We compare the performance of PCSM and PCSM (w/o Cali) with different frequency numbers $N^k$, as shown in Table. 8. PCSM always performs better than the counterpart model without spectral gates. The performance gap is particularly significant under lower frequency numbers (16, 32, and 64). This leads to the conclusion that the frequency calibration eliminates the dependency on a lot of frequencies, benefiting from the point-level flexible frequencies selection mechanism. Therefore, PCSM potentially performs better in some practical industry scenarios without sufficient spectral basis due to huge computational costs.

## A.5 Additional Visualization of Point-wise Frequency Preference

**Visualization of Spectral Gates for Samples with Different Resolutions.** Figure. 7 presents more visualization of spectral gates on Airfoil for single samples with varied resolutions. It could be observed that the frequency preference of each physical point remains consistent as the domain resolution varies. This demonstrates the resolution-agnostic characteristic of the Calibrated Spectral Transform.

**Visualization of Spectral Gates for Different Samples.** Figure. 8 shows the visualization of frequency preference for different samples on Darcy Flow. Although handling different input functions, the frequency calibration strategy is consistent for specific heads at specific layers. For example, the Head-1 of Layer-1 always enhances the high frequency of points at boundaries, and the Head-1 of Layer2 enhances the high frequency at regions with sharp status changes. This leads to the conclusion that the Multi-head Calibrated Spectral Mixer learns the modulated frequency calibration strategy for point-adaptive feature learning.

## A.6 Preliminary Background on Spectral Neural Mixer

**Spectral-based neural modules.** We note that the spectral processing has been extensively employed as basic neural modules for operator learning (Li et al., 2020; Chen et al., 2023; Tran et al., 2021), as well as general deep learning tasks (Guibas et al., 2021; Lee-Thorp et al., 2021). Unlike the classical neural modules such as CNN (LeCun et al., 1995), RNN (Chung et al., 2014), and Self-Attention mechanism (Vaswani, 2017) that directly mix features of different tokens or spatial locations in the spatial domain, these spectral-based neural modules learn features in the spectral domain.

Specifically, consider the input features $x \in \mathbb{R}^{N \times d}$, where $N$ is the number of tokens or spatial points and $d$ is the number of latent dimensions. The spectral-based neural module first transforms the features in the spatial domain into the spectral domain via spectral transform, after a few processes such as MLPs, the features are then transformed back to spatial domains. The spectral-based neural modules could be defined as follows:

$$\mathcal{F}_{\text{spectral}}^{\text{mixer}}(x) = \mathcal{T}^{-1} \circ \text{Project} \circ \mathcal{T}(x),$$

where $\mathcal{T}$ and $\mathcal{T}^{-1}$ represent the spectral transform and inverse spectral transform respectively. They could be instantiated in multiple ways, such as (inverse) Fourier Transform in Li et al. (2020) and

(inverse) Laplace-Beltrami Transform in Chen et al. (2023). Project is the point-wise neural modules such as MLPs.

**Definition of high-frequency and low-frequency features.** In the frequency domain, the features at different locations correspond to different frequencies. Consider the transformed spectral feature of $x \in \mathbb{R}^{N \times d}$ is $\hat{x} \in \mathbb{R}^{N^k \times d}$, i.e. $\hat{x} = \mathcal{T}(x)$. $\hat{x}$ consists of $N^k$ features, $\hat{x}_1, \hat{x}_2, ..., \hat{x}_{N^k}$, where each feature $\hat{x}_* \in \mathbb{R}^d$ corresponds to specific frequency.

In the Laplace-Beltrami Spectral Transform, we use the eigenfunctions of the Laplace-Beltrami Operator to transform the spatial signal $x \in \mathbb{R}^{N \times d}$ into spectral signals $\hat{x} \in \mathbb{R}^{N^k \times d}$. Consider the LBO eigenfunctions $\phi = [\phi_1, \phi_2, ..., \phi_{N^k}]$ where the $i$-th eigenfunction $\phi_i \in \mathbb{R}^{N \times 1}$ has the $i$-th lowest eigenvalue, $\phi_i$ could transform the spatial feature $x$ into the spectral feature at the $i$-th spectral feature $\hat{x}_i$ via matrix multiplication.

One property of the Laplace-Beltrami Spectral Transform is that the eigenfunctions associated with lower eigenvalues transform the signal to features of lower frequencies, whereas those with higher eigenvalues transform them to features of higher frequencies. Therefore, the frequency features $\hat{x}_* \in \mathbb{R}^d$ with lower indexes correspond to lower-frequency features, while those with high indexes correspond to higher-frequency features.

**The roles of high-frequency and low-frequency features in operator learning.** Intuitively, different frequency features help learn various aspects of function transformations for operator learning. As illustrated in Figure. 2, in the eigenfunctions with smaller eigenvalues (e.g. $\phi_{56}$, a lower frequency eigenfunction), the variation of importance across different spatial locations is smoother, which facilitates the learning of coarser-grained, slowly varying function transformations. Conversely, in the eigenfunction with larger eigenvalues (e.g. $\phi_{72}$, a higher frequency eigenfunction), the signal changes rapidly across different locations, aiding in learning finer-grained, faster-changing transformations.

As shown in Figure. 2, unlike conventional frequency eigenfunctions that are independent of the physical state of points, the introduced Point-Calibrated Spectral Transform calculates the features of different frequencies integrated with the physical state. Specifically, in Figure. 2 (b), regions with physical quantities changing rapidly could be automatically assigned stronger importance in the calculation of higher-frequency features. This could help them achieve enhanced prediction accuracy by integrating more local details.

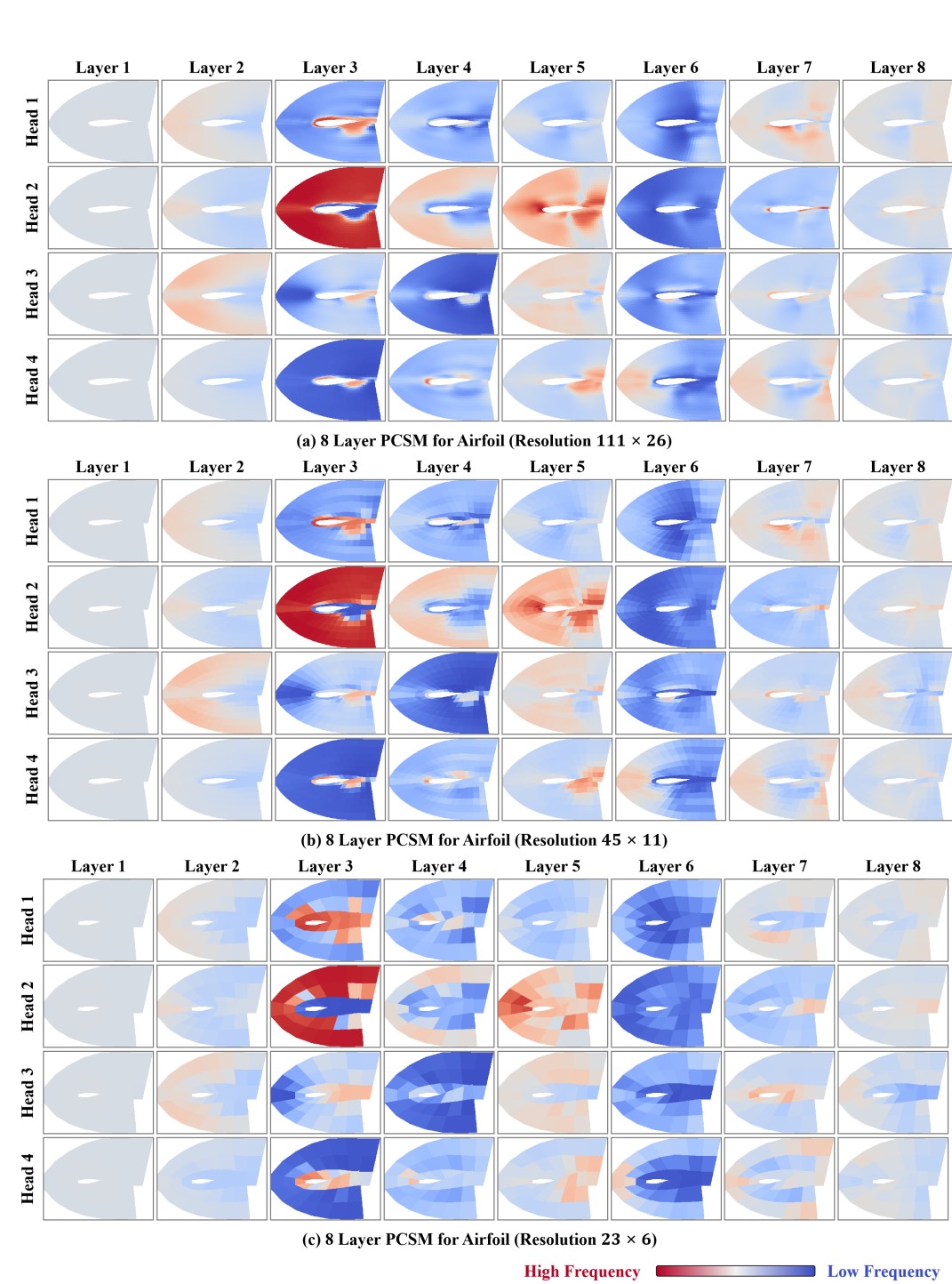

(a) 8 Layer PCSM for Airfoil (Resolution 111 × 26)

(b) 8 Layer PCSM for Airfoil (Resolution 45 × 11)

(c) 8 Layer PCSM for Airfoil (Resolution 23 × 6)

**High Frequency** **Low Frequency**

Figure 7: Visualization of point-wise frequency preference on Airfoil for samples with different resolutions.

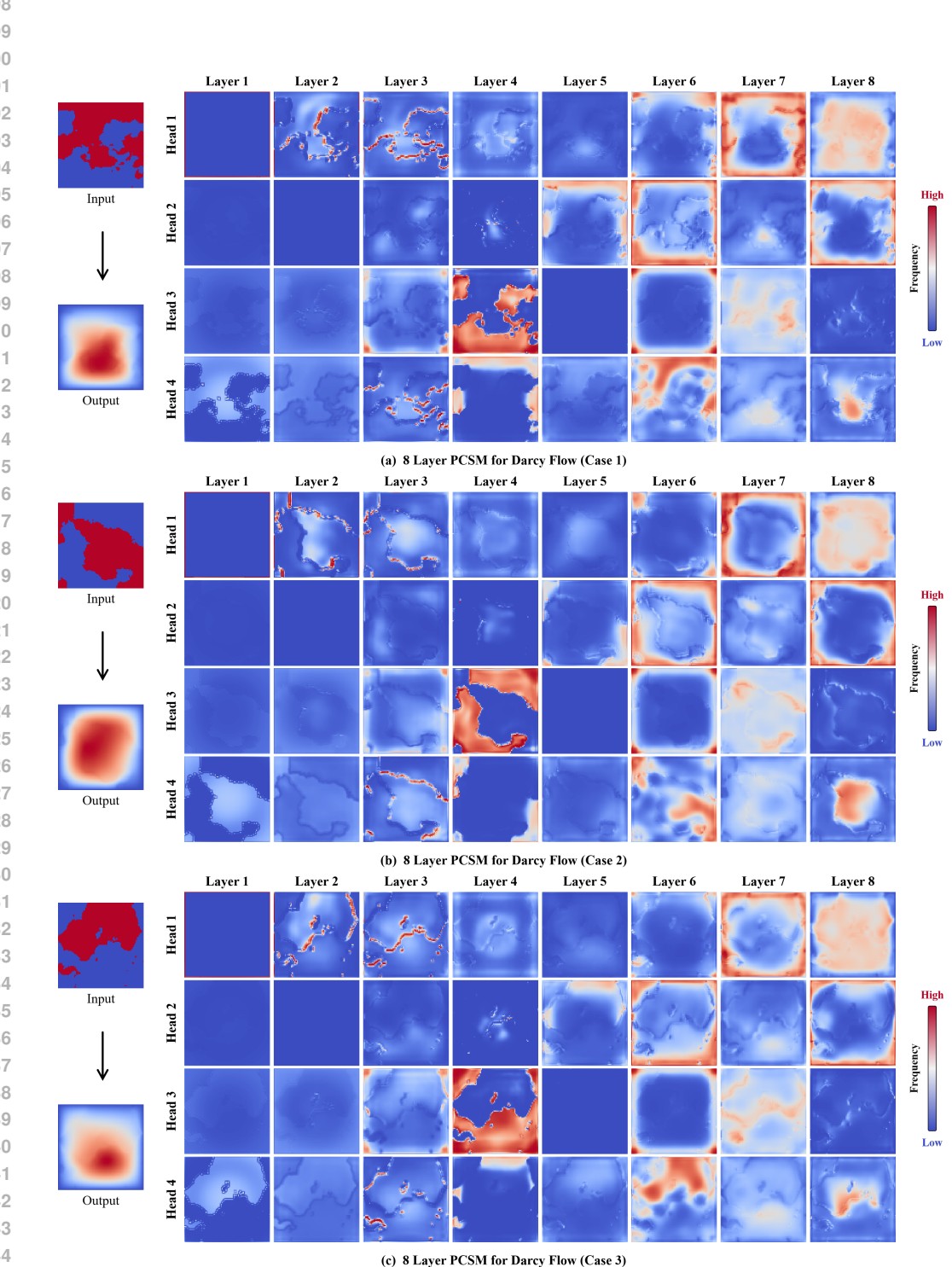

Figure 8: Visualization of point-wise frequency preference on Darcy Flow for different samples.

