# OpenReview forum: "Point-Calibrated Spectral Neural Operators"
_ICLR.cc/2025/Conference — Submitted to ICLR 2025_

### Official Review · Reviewer_pQPo · 2024-11-03

**Soundness:** 2
**Presentation:** 3
**Contribution:** 2
**Rating:** 5
**Confidence:** 4

**Summary:**

This paper presents a new neural architecture termed point-calibrated spectral neural operators for learning-based PDE solving.
The key idea is to adaptively compute a spectral gate to modulate/calibrate the fixed eigenfunctions (spectral basis) of the Laplace-Beltrami transform. Experiments on multiple PDE problems (with structured or unstructured meshes) collectively demonstrate the proposed methods outperform the state-of-the-art method (e.g., the Transolver) by a substantial margin, in terms of not only the approximation accuracy but also the sample efficiency and cross-resolution generalizability.

**Strengths:**

As I stated in the [Summary], the proposed method shows a clear improvement over the state-of-the-art in PDE fast solving. Also, the paper is in general clearly written and easy to digest.

**Weaknesses:**

However, the proposed method borrows significantly from the existing closely related works. Specifically, the spectral eigenfunctions (of the Laplace-Beltrami transform) are borrowed directly from an unpublished work [A]. And the basis form of the adaptive Gate functions (one of the key contributions claimed by authors) is widely employed in neural network literature (see, for instance, the squeeze-and-excitation network [B] proposed six years ago). For me, the main message claimed in this paper is that replacing the attention module by these aforementioned components in a classic transformer backbone leads to sizeable performance gain for neural operator learning. This is interesting, but may not be novel enough for ICLR publication.

Beside, I would not buy the explanation of "Pointwise Frequency Preference Learning" claimed by the authors. Indeed, the gate values are computed in a point-wise manner. However, it doesn't suggest it acts like a frequency selector. If one would implement an adaptive frequency selector, only one single scalar should be multiplied with each eigenfunction instead of the elementwise multiplication. IMO, the gate calibrates/modulates the eigenfunctions to make them more suitable to characterize the given data, but not in the form of frequency preference. After all, frequency can only be evaluated by a set of points instead of a single point (the uncertanty principle)

[A] Gengxiang Chen, Xu Liu, Qinglu Meng, Lu Chen, Changqing Liu, and Yingguang Li. Learning neural operators on riemannian manifolds. arXiv preprint arXiv:2302.08166, 2023.
[B] Hu, Jie, Li Shen, and Gang Sun. "Squeeze-and-excitation networks." Proceedings of the IEEE conference on computer vision and pattern recognition. 2018.

**Questions:**

See above

---

> ### Author Response · Authors · 2024-11-21
> **Response to Reviewer pQPo [Part 1]**
>
> We express our sincere gratitude to the reviewer for their constructive comments and valuable suggestions.
> The revised version of the manuscript has been updated, with the primary changes highlighted in blue.
> (All references of sections, tables, figures, and lines in the response correspond to those in the revised manuscript.)
> Some remaining content will be incorporated into the next version following further discussion.
> Below, we provide our responses to the questions raised.
>
> ## Weakness 1
>
> > W1: However, the proposed method borrows significantly from the existing closely related works. Specifically, the spectral eigenfunctions (of the Laplace-Beltrami transform) are borrowed directly from an unpublished work [A]. And the basis form of the adaptive Gate functions (one of the key contributions claimed by authors) is widely employed in neural network literature (see, for instance, the squeeze-and-excitation network [B] proposed six years ago). For me, the main message claimed in this paper is that replacing the attention module by these aforementioned components in a classic transformer backbone leads to sizeable performance gain for neural operator learning. This is interesting, but may not be novel enough for ICLR publication.
>
> We understand your concern about the innovation of our models, which incorporate the Laplace-Beltrami Spectral Transform and components in the classic transformer backbone.
> As stated in the original manuscript (line 107 and line 158), these components, employed in NORM [1], Transolver [2] and GNOT [3], form the foundation of our experiments, and we have properly acknowledged their contributions (proper citations in the original manuscript) and conducted fair comparisons beyond these components.
>
> As stated in the initial submission (line 78 - line 85), **our central contribution is the Point-Calibrated Spectral Mixer (PCSM)**, a fresh neural mixer based on the Point-Calibrated Spectral Transform. It innovatively combines the point-level flexibility of attention mixers and the domain-level continuty prior of spectral mixers.
>
> **(1) Performance superiority of PCSM**
>
> Firstly, **the fair comparison shows that the proposed PCSM module contributes significant performance gains**, beyond the existing modules (LBO Transform, classic Transformer components, etc.) used in previous methods.
> We extensively compare our model against two strong baselines, a robust spectral baseline (PCSM w/o Cali, based on NORM [1]) and a strong attention-based baseline (Transolver [2]).
> The only distinction between the two baselines and our model is the different mixer modules, while the backbone and optimization settings are identical.
> The results validate the superior performance of the proposed mixer (PCSM) compared to previous mixers (attention and fixed spectral mixers), in terms of the flexibility and the generalization capability.
>
> **(2) Technical novelty of PCSM**
>
> Furthermore, PCSM presents technical novelty compared to related works.
> - Regarding the Laplace-Beltrami Spectral Transform, we note that **PCSM is compatible with alternative spectral transforms**, including the Fourier Spectral Transform, as detailed in Section A.1.3 of the manuscript. We experiment with the Laplace-Beltrami Spectral Transform for its proficiency in handling irregular geometries.
> - Regarding the Neural Gate Mechanism, as discussed in Section 4 of the original manuscript, we acknowledge that the gate module has been extensively utilized for various purposes (Expert Selection in MoE [4], Information Controlling in GRU [5], etc.) in previous works. However, different from prior works, we explore the gate mechanism for Point-wise Frequency Calibration, which is **a novel design not studied before**. Additionally, our complemented theoretical demonstration in Section A.2.3 indicates that the introduced **neural gate module essentially acts like an implicit attention mechanism in PCSM (Remark A.9)**. As presented in Remark A.9, from the perspective of integral kernels, PCSM exquisitely integrates the attention module and spectral module. This demonstrates the technical innovation of PCSM and offers a theoretical basis for its superior performance.
>
> In summary, we believe that the proposed Point-Calibrated Spectral Mixer, although simple, indeed offers a fresh neural module simultaneously equipped with advantages of the popular spectral modules and attention modules, previously unaddressed.
> The novel module also possesses potential applicability in general deep learning tasks, such as visual data or time-series signal processing.
>
> **Reference**
> - [1] Learning Neural Operators on Riemannian Manifolds
> - [2] A Fast Transformer Solver for PDEs on General Geometries
> - [3] GNOT: A General Neural Operator Transformer for Operator Learning
> - [4] Outrageously large neural networks: The sparsely-gated mixture-of-experts layer
> - [5] On the properties of neural machine translation: Encoder-decoder approaches.

---

> ### Author Response · Authors · 2024-11-21
> **Response to Reviewer pQPo [Part 2]**
>
> ## Weakness 2
>
> > W2: Beside, I would not buy the explanation of "Pointwise Frequency Preference Learning" claimed by the authors. Indeed, the gate values are computed in a point-wise manner. However, it doesn't suggest it acts like a frequency selector. If one would implement an adaptive frequency selector, only one single scalar should be multiplied with each eigenfunction instead of the elementwise multiplication. IMO, the gate calibrates/modulates the eigenfunctions to make them more suitable to characterize the given data, but not in the form of frequency preference. After all, frequency can only be evaluated by a set of points instead of a single point (the uncertanty principle)
>
> We understand your opinion on the concept of "Pointwise Frequency Preference Learning".
> To address this, we provide a further explanation of the learning mechanism of PCSM in the following.
> These will be incorporated into our manuscript to make it more clear.
>
> **Explanation of Pointwise Frequency Preference Learning**
>
> - **Domain-Level Information Acquisition:**
> We recognize the principle that the frequency is related to a set of points rather than a single point. However, we wish to highlight that the point-wise gate module could acquire the domain-level information either in implicit or explicit manners. First, the point-wise gate module could adapt to the domain-level frequency eigenfunctions through the backpropagation-based optimization process, due to the frequency eigenfunctions being preset for each problem.
> Moreover, features at each point beyond the first layer could incorporate information from other points, enabling the prediction of frequency preference despite the pointwise processing.
>
> - **Neural Module Designs:** (1) Spectral gates prediction module. The softmax normalization facilitates the discrimination between different frequencies, guiding the module to learn how to allocate information selectively across frequencies, thereby promoting frequency preference learning. (2) Spectral calibration designs. We argue that both (case 1) multiplying each eigenfunction by a single scalar and (case 2) applying element-wise multiplication are viable designs aimed at enhancing feature learning in the spectral space. Both of them modify the spectral eigenfunctions with the learned frequency preference, to make them more suitable for specific problems. We have tested both approaches (in Table 7, Section A.4), where the global-level calibration (the last row) refers to case 1, and the point-level calibration (the first row) refers to case 2.  The results reveal clear performance gaps (4.38e-3 vs. 5.47e-3), which underscores the advantages of point-level calibration for feature learning in the spectral domain.
>
> - **Empirical Validation:** Our experiments (Section 3.4 and Section A.5) indicate that the spectral gates indeed exhibit a common understanding of point-level frequency preference, such as (1) preferring low-frequency features in most layers and (2) augmenting high-frequency features in boundary or rapidly changing regions. Additionally, as illustrated in Table 5, the learned frequency allocation principle across layers is consistent with the explicit frequency designs. These observations validate that the spectral gates actually learn the frequency preference of each physical point.
>
> In summary, even though the spectral gate values are calculated pointwise in each layer, they could learn and reflect the proper frequency preferences.
>
> **Further Insight into PCSM's Learning Mechanism**
>
> In addition to the intuitive "Frequency Preference Learning", the structure of PCSM can also be understood from the perspective of kernel functions.
> As suggested by Reviewer w1P9, our complemented theoretical analysis (Remark A.9 in Section A.2.3) demonstrates that PCSM essentially integrates the integral kernels of spectral and attention methods, providing further insight about its learning mechanism.

---

> ### Author Response · Authors · 2024-11-25
> **Friendly reminder: Have we addressed your concerns?**
>
> Dear Reviewer pQPo,
>
> We hope this message finds you well. We wish to extend our sincere gratitude to you for the expeditious and meticulous review you performed on our work.
>
> As we draw closer to the end of the discussion phase, we want to confirm that we have adequately addressed all of your concerns and questions about our work. We would be grateful if you could provide us with your further thoughts on our detailed rebuttal.
>
> If there are any additional questions or concerns that we have not yet fully addressed for a better score, please feel free to inform us. We will be happy to answer them as your feedback is crucial to us.
>
> Best regards,
>
> PCSM Authors.

---

> ### Author Response · Authors · 2024-12-01
> **Second Friendly Reminder: Awaiting Your Feedback on Our Rebuttal**
>
> Dear Reviewer pQPo,
>
> We hope this message finds you well. As we approach the delayed end of the discussion period, We would like to extend our gratitude again for your review of this work.
>
> We have submitted a rebuttal to address the points raised in your review and have been looking forward to your further thoughts. Could you please confirm whether you have had the opportunity to review our responses? We are eager to ensure that all your concerns have been adequately addressed. If there are any additional questions that we have not yet fully addressed for a better score, please feel free to inform us. We will be happy to answer them as your feedback is crucial to us.
>
> Thank you very much for your time and consideration.
>
> Best regards,
>
> PCSM Authors.

---

> ### Author Response · Authors · 2024-12-02
> **Third Friendly Reminder: Feedback on Rebuttal - Last Day of Discussion**
>
> Dear Reviewer pQPo,
>
> We hope this message finds you well. As today marks the last time (December 2nd at 11:59 pm AoE) of the discussion period, we are writing to gently inquire once more regarding your further thoughts on our rebuttal.
>
> We greatly appreciate your review and the comments provided, which have been instrumental in refining this work. We are eager to ensure that all aspects of your concern have been addressed, and we would be immensely grateful if you could share any additional comments on our rebuttal.
>
> We understand that your schedule is very busy, and we sincerely thank you for considering our request. Your feedback is crucial to us, and we look forward to hearing from you, even if it's just a brief note, to confirm that our responses have been received and reviewed.
>
> Thank you once again for your time and dedication to the review process.
>
> Best regards,
>
> PCSM Authors.

---

### Official Review · Reviewer_w1P9 · 2024-11-04

**Soundness:** 3
**Presentation:** 3
**Contribution:** 3
**Rating:** 5
**Confidence:** 5

**Summary:**

The paper proposes a Point-Calibrated Laplace-Beltrami Transform, which utilizes spatial information to assist the spectral selection of the spectral neural operator, effectively adapting to spatial variations in partial differential equation systems.

**Strengths:**

This method is very effective at adapting to spatial variations in systems of partial differential equations, and the organization and writing of the paper are quite good.

**Weaknesses:**

Minor typos:
The ‘Denotd’ in Figure 1.
Questions:
1. The Laplace-Beltrami Transform is an innovation inspired by the reference "Learning Neural Operators on Riemannian Manifolds." This paper merely introduces Pointwise Frequency Preference Learning based on that work. The introduction of auxiliary modules inevitably increases model parameters and computational complexity, but the paper does not provide ablation studies on parameters. Therefore, it is unclear whether the performance improvement is due to the increased parameters or the proposed algorithm.
2. The theoretical foundation is insufficient. The paper does not provide a unified kernel integral form as presented in [1]. Additionally, it does not prove that PCSM is equivalent to a learnable integral on Ω as demonstrated in [2].
3. In the Zero-shot Resolution Generalization experiment presented in the paper, the training resolution of 211 × 51 is greater than that of the test resolution. So, how would the model perform when generalizing to even larger resolutions?

[1] Neural operator: Learning maps between function spaces with applications to pdes.
[2] Transolver: A Fast Transformer Solver for PDEs on General Geometries.

**Questions:**

Minor typos:
The ‘Denotd’ in Figure 1.
Questions:
1. The Laplace-Beltrami Transform is an innovation inspired by the reference "Learning Neural Operators on Riemannian Manifolds." This paper merely introduces Pointwise Frequency Preference Learning based on that work. The introduction of auxiliary modules inevitably increases model parameters and computational complexity, but the paper does not provide ablation studies on parameters. Therefore, it is unclear whether the performance improvement is due to the increased parameters or the proposed algorithm.
2. The theoretical foundation is insufficient. The paper does not provide a unified kernel integral form as presented in [1]. Additionally, it does not prove that PCSM is equivalent to a learnable integral on Ω as demonstrated in [2].
3. In the Zero-shot Resolution Generalization experiment presented in the paper, the training resolution of 211 × 51 is greater than that of the test resolution. So, how would the model perform when generalizing to even larger resolutions?

[1] Neural operator: Learning maps between function spaces with applications to pdes.
[2] Transolver: A Fast Transformer Solver for PDEs on General Geometries.

---

> ### Author Response · Authors · 2024-11-21
> **Response to Reviewer w1P9 [Part 1]**
>
> We express our sincere gratitude to the reviewer for their constructive comments and valuable suggestions.
> The revised version of the manuscript has been updated, with the primary changes highlighted in blue.
> (All references of sections, tables, figures, and lines in the response correspond to those in the revised manuscript.)
> Some remaining content will be incorporated into the next version following further discussion.
> Below, we provide our responses to the questions raised.
>
> ## Weakness 1, Question 1
>
> > W1, Q1: The Laplace-Beltrami Transform is an innovation inspired by the reference "Learning Neural Operators on Riemannian Manifolds." This paper merely introduces Pointwise Frequency Preference Learning based on that work. The introduction of auxiliary modules inevitably increases model parameters and computational complexity, but the paper does not provide ablation studies on parameters. Therefore, it is unclear whether the performance improvement is due to the increased parameters or the proposed algorithm.
>
> We acknowledge that the introduced Point-Calibrated module inevitably increases the number of parameters.
> However, we claim that the performance gains by Point-Calibrated module are from the effective combination of the point-level flexibility (like attention-based methods) and the spectral priors (like previous spectral-based methods), rather than merely due to the augmented parameter count.
> To illustrate this, we provide the following complemented results.
>
> 1. PCSM could significantly outperform the strong baselines (PCSM w/o Cali based on NORM [1], and Transolver [2]), even when using fewer layers (only 4 layers, and thus fewer parameters).
> The comparison on Darcy Flow is shown in the table below:
>
> | Model | Layer Number | Parameter Amount | L2 Error |
> |----|----|----|----|
> | Transolver | 8  | 2,835,649   | 5.24e-3  |
> | PCSM (w/o Cali) | 8  | 617,665  | 5.31e-3  |
> | PCSM  | 8  | 767,169  | 4.38e-3 |
> | PCSM  | 4  | 408,737   | 4.71e-3  |
>
> 2. Compared to attention-based architectures, the increment in the number of parameters of the Point-Calibrated module remains relatively modest.
> The following table compares the parameter amount across different problems:
>
> | Problem | PCSM (w/o Cali) | PCSM | Transolver |
> |----|----|----|----|
> | Darcy Flow  | 617,665   | 767,169   | 2,835,649  |
> | Airfoil | 601,537   | 751,041   | 2,819,521  |
> | Plasticity    | 635,204   | 784,708   | 2,853,188  |
> | Navier-Stokes  | 2,325,313 | 2,868,545 | 11,240,769 |
>
> Therefore, we believe that the introduced Point-Calibrated Spectral Mixer, despite its simplicity, indeed contributes substantial performance gains in operator learning.
> Additionally, it innovatively integrates the strengths of spectral models and attention models, thereby offering significant technical novelty.
>
> **Reference**
> - [1] Learning neural operators on riemannian manifolds
> - [2] A Fast Transformer Solver for PDEs on General Geometries
>
> ## Weakness 2, Question 2
>
> > W2, Q2: The theoretical foundation is insufficient. The paper does not provide a unified kernel integral form as presented in [1]. Additionally, it does not prove that PCSM is equivalent to a learnable integral on Ω as demonstrated in [2].
>
> We include the theoretical foundation in Section A.2 of the revised manuscript.
>
> In addition to deriving that **PCSM is equivalent to a learnable integral on $\Omega$** (Theorem A.8, Equation 38) as you suggested, we also complement **a formal definition of the Point-Calibrated Spectral Transform** (Definition A.2, Equations 20 and 21).
>
> Furthermore, from the perspective of a kernel integral operator, we demonstrate that **PCSM unifies the kernel functions of attention-based and spectral-based models** (Remark A.9, Equation 39).
> If the spectral component is removed, PCSM reduces to a linear attention mechanism.
> Conversely, if the component of attention methods is excluded, PCSM becomes a classical spectral mixer.
> This is consistent with the empirical observation that PCSM combines the flexibility of attention mechanisms with the spectral prior of spectral methods.
> This further highlights the technical innovation of PCSM and offers a theoretical basis for its superior performance.

---

> ### Author Response · Authors · 2024-11-21
> **Response to Reviewer w1P9 [Part 2]**
>
> ## Weakness 3, Question 3
>
> > W3, Q3: In the Zero-shot Resolution Generalization experiment presented in the paper, the training resolution of 211 × 51 is greater than that of the test resolution. So, how would the model perform when generalizing to even larger resolutions?
>
> To compare the performance when generalizing to larger resolution, we train the models on resolution $111 \times 26$ and evaluate them on resolution $221 \times 51$ (unseen higher resolution) and $45 \times 11$ (unseen lower resolution) on Airfoil problem.
> The results are summarized in the table below.
>
> | Model | $111 \times 26$ (Train) | $221 \times 51$ (Higher-Resolution Transfer) | $45 \times 11$ (Lower-Resolution Transfer) |
> |----|----|----|----|
> | Transolver | 4.50e-3 | 6.95e-2 | 1.22e-1 |
> | PCSM (w/o Cali) | 5.37e-3 | 1.95e-2 | 5.84e-2 |
> | PCSM | **4.17e-3** | **1.57e-2** | **3.85e-2** |
>
> Although the attention-based model (Transolver) presents excellent performance at the training resolution of $111 \times 26$, its performance drops a lot when transferred to both the higher ($221 \times 51$) and lower ($45 \times 11$) unseen resolutions, where it underperforms compared to the spectral-based method (PCSM w/o Cali).
>
> In contrast, PCSM not only matches the impressive performance of the attention-based method (Transolver) at the training resolution but also exhibits preferred generalization ability to the unseen resolutions, maintaining superiority over Transolver in both higher and lower resolution settings.
> This suggests that PCSM effectively combines the benefits of high accuracy at the training domain (like the attention-based baseline, Transolver) with strong generalization capabilities (like the spectral-based baseline, PCSM w/o Cali).

---

> ### Author Response · Authors · 2024-11-25
> **Friendly reminder: Have we addressed your concerns?**
>
> Dear Reviewer w1P9,
>
> We hope this message finds you well. We wish to extend our sincere gratitude to you for the expeditious and meticulous review you performed on our work.
>
> As we draw closer to the end of the discussion phase, we want to confirm that we have adequately addressed all of your concerns and questions about our work. We would be grateful if you could provide us with your further thoughts on our detailed rebuttal.
>
> If there are any additional questions or concerns that we have not yet fully addressed for a better score, please feel free to inform us. We will be happy to answer them as your feedback is crucial to us.
>
> Best regards,
>
> PCSM Authors.

---

> ### Author Response · Authors · 2024-12-01
> **Second Friendly Reminder: Awaiting Your Feedback on Our Rebuttal**
>
> Dear Reviewer w1P9,
>
> We hope this message finds you well. As we approach the delayed end of the discussion period, We would like to extend our gratitude again for your review of this work.
>
> We have submitted a rebuttal to address the points raised in your review and have been looking forward to your further thoughts. Could you please confirm whether you have had the opportunity to review our responses? We are eager to ensure that all your concerns have been adequately addressed. If there are any additional questions that we have not yet fully addressed for a better score, please feel free to inform us. We will be happy to answer them as your feedback is crucial to us.
>
> Thank you very much for your time and consideration.
>
> Best regards,
>
> PCSM Authors.

---

> ### Author Response · Authors · 2024-12-02
> **Third Friendly Reminder: Feedback on Rebuttal - Last Day of Discussion**
>
> Dear Reviewer w1P9,
>
> We hope this message finds you well. As today marks the last time (December 2nd at 11:59 pm AoE) of the discussion period, we are writing to gently inquire once more regarding your further thoughts on our rebuttal.
>
> We greatly appreciate your review and the comments provided, which have been instrumental in refining this work. We are eager to ensure that all aspects of your concern have been addressed, and we would be immensely grateful if you could share any additional comments on our rebuttal.
>
> We understand that your schedule is very busy, and we sincerely thank you for considering our request. Your feedback is crucial to us, and we look forward to hearing from you, even if it's just a brief note, to confirm that our responses have been received and reviewed.
>
> Thank you once again for your time and dedication to the review process.
>
> Best regards,
>
> PCSM Authors.

---

### Official Review · Reviewer_6qMY · 2024-11-04

**Soundness:** 3
**Presentation:** 3
**Contribution:** 3
**Rating:** 6
**Confidence:** 3

**Summary:**

The authors have addressed some of my concerns, but I still have questions regarding the low-frequency and high-frequency aspects. I am willing to consider increasing my score.

**Strengths:**

Good organization and structure of the paper, along with solid experimental analysis.

**Weaknesses:**

The authors have addressed some of my concerns, but I still have questions regarding the low-frequency and high-frequency aspects.

**Questions:**

The authors have addressed some of my concerns, but I still have questions regarding the low-frequency and high-frequency aspects.

---

> ### Author Response · Authors · 2024-11-21
> **Response to Reviewer 6qMY [Part 1]**
>
> We express our sincere gratitude to the reviewer for their constructive comments and valuable suggestions.
> The revised version of the manuscript has been updated, with the primary changes highlighted in blue.
> (All references of sections, tables, figures, and lines in the response correspond to those in the revised manuscript.)
> Some remaining content will be incorporated into the next version following further discussion.
> Below, we provide our responses to the questions raised.
>
> ## Weakness 1, Question 1
>
> > W1: The authors did not provide a detailed explanation of how low frequency and high frequency are defined, as well as their respective roles. \
> > Q1: A detailed explanation of how low frequency and high frequency are defined, along with their respective roles, should be included.
>
> Thank you for your valuable suggestions regarding the explanation of spectral frequencies.
> To better illustrate our work, we provide additional summaries concerning low-frequency and high-frequency features in the following.
> These will be incorporated into our manuscript.
>
> **(1) Definition of Low-Frequency and High-Frequency**
>
> In the frequency domain, we can relatively assess the frequency levels at different positions.
> Generally, there is no absolute boundary to distinguish between low and high frequencies unless specifically stated.
> In the Laplace-Beltrami Spectral Transform, the eigenfunctions of the Laplace-Beltrami Operator could transform the spatial signal into signals of varying frequencies.
> The eigenfunctions associated with lower eigenvalues transform the signal to features of lower frequencies, whereas those with higher eigenvalues transform them to features of higher frequencies, as previously presented in [1,2].
>
> **(2) Roles of Low-Frequency and High-Frequency**
>
> Intuitively, different frequency features help in learning different aspects of function transformations for operator learning.
> As illustrated in Figure 2, in the eigenfunctions with smaller eigenvalues (e.g. $\phi_{56}$, a lower frequency eigenfunction), the variation of importance across different spatial locations is smoother, which facilitates the learning of smooth, slowly varying function transformations.
> Conversely, in the eigenfunction with larger eigenvalues (e.g. $\phi_{72}$, a higher frequency eigenfunction), the signal changes rapidly across different locations, aiding in learning finer-grained, faster-changing transformations.
>
> As shown in Figure 2, unlike conventional frequency eigenfunctions that are independent of the physical state of points, the introduced Point-Calibrated Spectral Transform calculates the features of different frequencies integrated with the physical state.
> Specifically, in Figure 2(b), regions where physical quantities change rapidly could be automatically assigned stronger importance in the calculation of higher-frequency features.
> This could help them achieve enhanced prediction accuracy.
>
> **Reference**
> - [1] Discrete Differential-Geometry Operators for Triangulated 2-Manifolds
> - [2] A Laplacian for Nonmanifold Triangle Meshes

---

> ### Author Response · Authors · 2024-11-21
> **Response to Reviewer 6qMY [Part 2]**
>
> ## Weaknesses 2, 3 and Questions 2, 3
>
> > W2: The authors' introduction of the POINT-CALIBRATED SPECTRAL NEURAL OPERATOR is simplistic and feels somewhat cursory. \
> > Q2: The authors should provide a more detailed introduction to the POINT-CALIBRATED SPECTRAL NEURAL OPERATOR and include a textual comparison with other methods. \
> > W3: The introduction to neural operators is too brief, appearing only in section 2.1.1, which creates significant obstacles for later reading. \
> > Q3: The authors should expand on the introduction to neural operators and include transitional statements leading to section 2.2, POINT-CALIBRATED SPECTRAL NEURAL OPERATOR.
>
> Thank you for your valuable suggestions regarding the presentation of the introduced Point-Calibrated Spectral Mixer (Weakness 2 and Question 2) and the background on neural operators (Weakness 3 and Question 3). To address your concerns and enhance clarity, we have made the following improvements:
>
> **Regarding the Introduction to Point-Calibrated Spectral Neural Operator (Weakness 2 and Question 2)**
> - (1) To aid comprehension, we have added a formal definition of the Point-Calibrated Spectral Transform in Section A.2.1.
> - (2) For a more comprehensive presentation (not "cursory") of the Point-Calibrated Spectral Mixer (PCSM), we detail the derivation process from the integral neural operator [1] to the Point-Calibrated Spectral Neural Operator in Equation 38 (Section A.2.3).
> - (3) Moreover, beyond the intuitive explanation of frequency preference learning, we offer a more theoretical comparison (the "textual comparison") between PCSM and prior models in Remark A.9 (Section A.2.3), demonstrating that PCSM implicitly integrates previous linear attention mechanism and the fixed spectral models.
>
> **Regarding the Introduction to Neural Operators (Weakness 3 and Question 3)**
> - (1) To improve understanding and readability, we have included an introduction to the classic form of neural operators in Section A.2.2, and introduced the connection between the transform-based neural operators and the classic neural operator in Section 2.1.2.
> - (2) Additionally, we have provided the formal definition of the Laplace-Beltrami Spectral Transform used in our work in Section A.2.1.
>
> We believe these modifications will help readers better understand our methodology. We will further refine our presentation to ensure it is as accessible and clear as possible.
>
> **Reference**
> - [1] Neural operator: Learning maps between function spaces with applications to pdes

---

> > ### Comment · Reviewer_6qMY · 2024-11-27
> > **Thank you for providing a rebuttal**
> >
> > The authors have addressed some of my concerns, but I still have questions regarding the low-frequency and high-frequency aspects. I am willing to consider increasing my score.

---

> > > ### Author Response · Authors · 2024-11-28
> > > **Thank you for the response**
> > >
> > > We are very grateful for your response and for recognizing our rebuttal.
> > >
> > > In response to your remaining question concerning low-frequency and high-frequency components, we have added the introduction to the spectral mixer modules in Section A.6 of the revised manuscript. This section includes (1) a review of spectral neural mixers introduced in prior works, (2) the detailed definition and calculation for features at varying frequencies, (3) and the distinct roles of different frequency features. We hope this enhanced explanation will address your questions.
> > >
> > > If you have any further questions, please feel free to ask. We are more than willing to answer them.

---

> > > > ### Author Response · Authors · 2024-11-28
> > > >
> > > > For your convenience, we have placed the added introduction to the spectral mixer modules in the following:
> > > >
> > > > **Background of spectral-based neural modules.**
> > > >
> > > > We emphasize that spectral processing has been extensively employed as basic neural modules for operator learning [1,2,3,4], as well as general deep learning tasks [5,6].
> > > > Unlike the classical neural modules such as CNN, RNN, and Self-Attention mechanism that directly mix features of different tokens or spatial locations in the spatial domain, these spectral-based neural modules learn features in the spectral domain.
> > > >
> > > > Specifically, consider the input features $x \in \mathbb{R}^{N \times d}$, where $N$ is the number of tokens or spatial points and $d$ is the number of latent dimensions.
> > > > The spectral-based neural module first transforms the features in the spatial domain into the spectral domain via spectral transform, after a few processes such as MLPs, the features are then transformed back to spatial domains.
> > > > The spectral-based neural modules could be defined as follows:
> > > > $$
> > > > \mathcal{F}_{\text{spectral}}^{mixer}(x) =  \mathcal{T}^{-1} \circ \text{Project} \circ \mathcal{T}(x),
> > > > $$
> > > > where $\mathcal{T}$ and $\mathcal{T}^{-1}$ represent the spectral transform and inverse spectral transform respectively.
> > > > They could be instantiated in multiple ways, such as the (inverse) Fourier Transform in [1] and the (inverse) Laplace-Beltrami Transform in [2].
> > > > $\text{Project}$ is the point-wise neural modules such as MLPs.
> > > >
> > > > **Definition of high-frequency and low-frequency features.**
> > > >
> > > > In the frequency domain, the features at different locations correspond to different frequencies.
> > > > Consider the transformed spectral feature of $x \in \mathbb{R}^{N \times d}$ is $\hat{x} \in \mathbb{R}^{N^k \times d}$, i.e. $\hat{x}=\mathcal{T}(x)$.
> > > > $\hat{x}$ consists of $N^k$ features, $\hat{x}_ 1, \hat{x}_ 2, ...,\hat{x}_ {N^k}$, where each feature $\hat{x}_* \in \mathbb{R}^d$ corresponds to specific frequency.
> > > >
> > > > In the Laplace-Beltrami Spectral Transform, we use the eigenfunctions of the Laplace-Beltrami Operator to transform the spatial signal $x \in \mathbb{R}^{N \times d}$ into spectral signals $\hat{x} \in \mathbb{R}^{N^k \times d}$.
> > > > Consider the LBO eigenfunctions $\phi=[\phi_1, \phi_2, ..., \phi_{N^k}]$ where the $i$-th eigenfunction $\phi_i \in \mathbb{R}^{N \times 1}$ has the $i$-th lowest eigenvalue, $\phi_i$ could transform the spatial feature $x$ into the spectral feature at the $i$-th spectral feature $\hat{x}_i$ via matrix multiplication.
> > > >
> > > > One property of the Laplace-Beltrami Spectral Transform is that the eigenfunctions associated with lower eigenvalues transform the signal to features of lower frequencies, whereas those with higher eigenvalues transform them to features of higher frequencies.
> > > > Therefore, the frequency features $\hat{x}_* \in \mathbb{R}^d$ with lower indexes correspond to lower-frequency features, while those with high indexes correspond to higher-frequency features.
> > > >
> > > > **The roles of high-frequency and low-frequency features in operator learning.**
> > > >
> > > > Intuitively, different frequency features help learn various aspects of function transformations for operator learning.
> > > > As illustrated in Figure 2, in the eigenfunctions with smaller eigenvalues (e.g. $\phi_{56}$, a lower frequency eigenfunction), the variation of importance across different spatial locations is smoother, which facilitates the learning of smooth, slowly varying function transformations.
> > > > Conversely, in the eigenfunction with larger eigenvalues (e.g. $\phi_{72}$, a higher frequency eigenfunction), the signal changes rapidly across different locations, aiding in learning finer-grained, faster-changing transformations.
> > > >
> > > > As shown in Figure 2, unlike conventional frequency eigenfunctions that are independent of the physical state of points, the introduced Point-Calibrated Spectral Transform calculates the features of different frequencies integrated with the physical state. Specifically, in Figure 2(b), regions with physical quantities changing rapidly could be automatically assigned stronger importance in the calculation of higher-frequency features. This could help them achieve enhanced prediction accuracy.
> > > >
> > > > **Reference**
> > > > - [1] Fourier Neural Operator for Parametric Partial Differential Equations
> > > > - [2] Learning Neural Operators on Riemannian Manifolds
> > > > - [3] Factorized Fourier Neural Operators
> > > > - [4] U-NO: U-shaped Neural Operators
> > > > - [5] Adaptive Fourier Neural Operators: Efficient Token Mixers for Transformers
> > > > - [6] FNet: Mixing Tokens with Fourier Transforms

---

> ### Author Response · Authors · 2024-11-25
> **Friendly reminder: Have we addressed your concerns?**
>
> Dear Reviewer 6qMY,
>
> We hope this message finds you well. We wish to extend our sincere gratitude to you for the expeditious and meticulous review you performed on our work.
>
> As we draw closer to the end of the discussion phase, we want to confirm that we have adequately addressed all of your concerns and questions about our work. We would be grateful if you could provide us with your further thoughts on our detailed rebuttal.
>
> If there are any additional questions or concerns that we have not yet fully addressed for a better score, please feel free to inform us. We will be happy to answer them as your feedback is crucial to us.
>
> Best regards,
>
> PCSM Authors.

---

### Official Review · Reviewer_Eei4 · 2024-11-18

**Soundness:** 3
**Presentation:** 3
**Contribution:** 3
**Rating:** 6
**Confidence:** 3

**Summary:**

The paper introduces the Point-Calibrated Spectral Neural Operator. This new approach combines the flexibility of attention-based methods and the spectral continuity constraints from spectral-based neural operators.

**Strengths:**

This paper introduces a novel hybrid approach by integrating spectral methods with point-wise adaptability, which is an advancement in the field of neural PDE solvers. It bridges the gap between spectral continuity and spatial flexibility, potentially positioning it as a unique contribution to both operator learning and spectral-based neural networks.

The paper demonstrates a deep understanding of spectral analysis and neural operator design, incorporating the Laplace-Beltrami Transform to handle irregular domains and non-uniform meshes. The theoretical basis for using point-wise frequency preferences and neural gates to calibrate spectral features seems well-founded and addresses specific challenges of both spectral and attention-based methods.

The experiments cover a range of structured and unstructured PDE problems, offering quantitative and qualitative results that consistently show the advantages of PCSM over baselines. The paper explores zero-shot resolution generalization and limited training data scenarios, demonstrating PCSM’s robustness, adaptability, and reduced dependence on extensive data or frequency inputs.

**Weaknesses:**

1. The methodology, while innovative, may appear complex to readers unfamiliar with spectral methods, neural operators, or Laplace-Beltrami transforms. This could limit accessibility for a broader audience within ICLR, where general machine learning applications dominate. Some sections might benefit from clearer explanations or illustrative examples, especially in the methodology, to help general readers understand the motivation behind certain design choices.

2. Although the PCSM model integrates spatial adaptability, its reliance on spectral transforms might make it challenging to apply in domains where calculating or interpreting spectral features (e.g., LBO eigenfunctions) is difficult or computationally expensive. This approach may also limit the model’s use in non-spectral applications unless the paper provides guidelines on extending or adapting the point-calibrated spectral mechanism to more generalized feature spaces.

3. The experiments utilize computationally demanding resources (like the A100 GPU), which could suggest that PCSM is resource-intensive, particularly for calculating point-wise frequency preferences and training on large irregular domains. Although results show reduced dependence on high-frequency inputs, the model still involves complex frequency selection and spectral processing steps, which might require optimization for real-world applications.

4.  While the paper’s experiments are extensive and well-rounded, they are conducted on benchmark problems. Including a real-world PDE application, or additional industry-related scenarios, could strengthen the claim of broader applicability. Demonstrating PCSM’s utility outside synthetic PDE problems, in genuinely complex physical systems or industrial settings, would bolster its practical significance.

**Questions:**

1. Comparative Analysis: The paper presents a strong case for the PCSM approach over fixed spectral methods. Could further analysis be provided to illustrate specific PDE scenarios where point-calibration particularly excels compared to the fixed approach?

2. Computational Efficiency: Given the reported need for A100 GPUs, are there any considerations for deploying PCSM in resource-constrained environments? Could simplified versions of PCSM be implemented without significant accuracy loss?

3. Real-World Application Scenarios: Has PCSM been tested on any real-world data or practical engineering problems? Including results from real-world applications (e.g., fluid dynamics in engineering contexts) could add to the paper’s practical significance.

4. Sensitivity to Frequency Parameters: How sensitive is PCSM’s performance to changes in the number of frequency modes (Nk)? Are there guidelines for choosing this parameter in cases where computational resources are limited?

5. Ablation on Multi-Head Mechanism: The multi-head mechanism is mentioned as an enhancement to the spectral mixer. Have the authors conducted any ablation studies to understand how varying the number of heads affects performance?

6. Explainability of Point-Wise Frequency Preferences: Have the authors considered using interpretability techniques to explain the model’s learned point-wise frequency preferences? For instance, do specific frequencies correspond to particular physical phenomena in the PDE domains?

7. Future Extensions of Point-Calibrated Spectral Transform: The paper suggests broader applicability for time-series and computer vision tasks. Could the authors provide some insights into how the Point-Calibrated Spectral Transform might be adapted for non-spectral data, such as images?

Minor comment:
1. Denotd should be Denoted in Figure 1: “Denotd as 'PCSM (w/o Cali)'”​

---

> ### Author Response · Authors · 2024-11-21
> **Response to Reviewer Eei4 [Part 1]**
>
> We express our sincere gratitude to the reviewer for their constructive comments and valuable suggestions.
> The revised version of the manuscript has been updated, with the primary changes highlighted in blue.
> (All references of sections, tables, figures, and lines in the response correspond to those in the revised manuscript.)
> Some remaining content will be incorporated into the next version following further discussion.
> Below, we provide our responses to the questions raised.
>
> ## Weakness 1
>
> > W1: The methodology, while innovative, may appear complex to readers unfamiliar with spectral methods, neural operators, or Laplace-Beltrami transforms. This could limit accessibility for a broader audience within ICLR, where general machine learning applications dominate. Some sections might benefit from clearer explanations or illustrative examples, especially in the methodology, to help general readers understand the motivation behind certain design choices.
>
> Thank you for your suggestions regarding the presentation. As our response to Reviewer 6qMY, we will further refine our presentation in the following aspects:
>
> **(1) Introduction to Neural Operators**
> - To enhance clarity and readability, we have introduced the classical form of neural operators in Section A.2.2 and explained the relationship between transform-based neural operators and the classical neural operator in Section 2.1.2.
> - We have provided a formal definition of the Laplace-Beltrami Spectral Transform used in our work in Section A.2.1.
>
> **(2) Introduction to Point-Calibrated Spectral Neural Operator**
> - To aid understanding, we have added a formal definition of the Point-Calibrated Spectral Transform in Section A.2.1.
> - For a more comprehensive presentation of the Point-Calibrated Spectral Mixer (PCSM), we have detailed the derivation steps from the integral neural operator [1] to the Point-Calibrated Spectral Neural Operator in Equation 38 (Section A.2.3).
> - Beyond the intuitive explanation of frequency preference learning, we provide a theoretical comparison between PCSM and prior models in Remark A.9 (Section A.2.3), introducing how PCSM integrates previous linear attention mechanisms and fixed spectral models.
>
> **(3) Explanation about Spectral Processing**
> - To help understand the spectral processing for readers unfamiliar with spectral transform, we will include a brief introduction to the definitions of low-frequency and high-frequency features, as well as their distinct roles in function approximation for operator learning.
>
> We believe these modifications will improve the clarity and accessibility of our methodology. We will continue to refine our presentation to ensure it is as clear as possible.
>
> **Reference**
> - [1] Neural operator: Learning maps between function spaces with applications to pdes
>
>
> ## Weakness 2
>
> > W2: Although the PCSM model integrates spatial adaptability, its reliance on spectral transforms might make it challenging to apply in domains where calculating or interpreting spectral features (e.g., LBO eigenfunctions) is difficult or computationally expensive. This approach may also limit the model’s use in non-spectral applications unless the paper provides guidelines on extending or adapting the point-calibrated spectral mechanism to more generalized feature spaces.
>
> We acknowledge the computational cost associated with the calculation of spectral features such as eigenfunctions. This is indeed a common limitation of most spectral-based methods. However, we would like to emphasize the following points:
>
> (1) **For most PDEs, the computational cost remains manageable even for large domains**, because only a small number of frequencies are necessary to approximate solution functions over large domains. This has been extensively confirmed in prior works [1,2,3]. For example, as shown in FNO [1], only 64 frequency features are required for the approximation of functions defined on 4096 spatial points for the Navier-Stokes problem.
>
> (2) Compared to previous fixed spectral methods, **PCSM requires less computational cost on spectral features**. This is benefited from its weaker dependency on the number of spectral frequencies, as detailed in the response to Question 4.
>
> (3) Although this work primarily focuses on the spectral scenario (PDE solving), in response to Question 7, we present that **PCSM could be extended to handle these non-spectral data such as images** without introducing significant computational cost.
>
> In summary, while the computational cost of spectral transforms exists, the efficiency and flexibility of PCSM could mitigate this issue, making it a robust and versatile approach.
>
> **Reference**
> - [1] Fourier Neural Operator for Parametric Partial Differential Equations
> - [2] Transform Once Efficient Operator Learning in Frequency Domain
> - [3] Learning Neural Operators on Riemannian Manifolds

---

> ### Author Response · Authors · 2024-11-21
> **Response to Reviewer Eei4 [Part 2]**
>
> ## Weakness 3, Question 2
>
> > W3: The experiments utilize computationally demanding resources (like the A100 GPU), which could suggest that PCSM is resource-intensive, particularly for calculating point-wise frequency preferences and training on large irregular domains. Although results show reduced dependence on high-frequency inputs, the model still involves complex frequency selection and spectral processing steps, which might require optimization for real-world applications. \
> > Q2: Computational Efficiency: Given the reported need for A100 GPUs, are there any considerations for deploying PCSM in resource-constrained environments? Could simplified versions of PCSM be implemented without significant accuracy loss?
>
> We note that the A100 device is not necessary for training the proposed PCSM.
> The device clarification in Section A.3.1 means a single A100 device is enough (not necessary) for all experiments (not only running PCSM, but also running baselines, and ablation studies).
> Further testing has demonstrated that "a single RTX 3090 device" can run PCSM for most problems, even those with a large number of points, such as the Airfoil problem with 11,271 points.
> We will include this information in the manuscript to prevent misunderstanding.
>
> PCSM retains the efficiency compared to previous neural operators:
> 1. Compared to traditional fixed spectral methods, **the additional computational cost of PCSM is minimal**, including only the point-wise spectral gate module and the element-wise matrix multiplication. These operations introduce a modest increase in model parameters and computation time compared to attention-based neural operators like Transolver [1].
> 2. Additionally, as noted in our response to Weakness 2, **the computational expense associated with spectral processing and point-wise frequency preference prediction remains manageable for PCSM even for large domains**. This is because most PDE problems do not require a high number of frequencies, even for larger domains. What's more, PCSM is insensitive to the number of frequencies, which allows more efficient spectral processing with fewer frequencies in practical scenarios.
> 3. To validate the limited cost introduced by Point-Calibrated Spectral Mixer, we compare the parameter amount and inference time of different methods on the Airfoil problem. The running time is evaluated on a single RTX 3090 device. The results are summarized in the table below. It could be observed that the increase in parameters and computational time for PCSM is indeed minimal. These results will be incorporated into our manuscript.
>
> | Model         | Parameter Count | Average Inference Time Per Sample |
> |---------------|-----------------|----------------|
> | PCSM (w/o Cali) | 601,537       | 14.8 ms        |
> | PCSM           | 751,041        | 17.4 ms        |
> | Transolver     | 2,819,521      | 30.9 ms        |
>
>
> **Reference**
> - [1] Transolver: A Fast Transformer Solver for PDEs on General Geometries
>
>
> ## Weakness 4, Question 3
>
> > W4: While the paper’s experiments are extensive and well-rounded, they are conducted on benchmark problems. Including a real-world PDE application, or additional industry-related scenarios, could strengthen the claim of broader applicability. Demonstrating PCSM’s utility outside synthetic PDE problems, in genuinely complex physical systems or industrial settings, would bolster its practical significance. \
> > Q3: Real-World Application Scenarios: Has PCSM been tested on any real-world data or practical engineering problems? Including results from real-world applications (e.g., fluid dynamics in engineering contexts) could add to the paper’s practical significance.
>
> Thank you for your valuable suggestions.
>
> We would like to emphasize that our current experimental results have included several problems derived from real-world or industry-related scenarios.
> For example, the Composite problem, as provided by [1], seeks to predict the deformation fields of Carbon Fiber Reinforced Polymer (CFRP) under high-temperature conditions.
> This problem directly stems from the production process of the air intake parts of jets used in the aerospace industry.
> Additionally, the Blood Flow problem focuses on simulating the hemodynamics of the human thoracic aorta, the largest artery responsible for delivering oxygen and nutrient-rich blood.
> This is another real-world PDE problem with practical significance.
>
> We acknowledge the importance of expanding our scope to include more real-world PDE applications.
> Therefore, we plan to further validate the performance of PCSM on more complex and diverse problems in the future.
>
> **Reference**
> - [1] Learning Neural Operators on Riemannian Manifolds

---

> ### Author Response · Authors · 2024-11-21
> **Response to Reviewer Eei4 [Part 3]**
>
> ## Question 1
>
> > Q1: Comparative Analysis: The paper presents a strong case for the PCSM approach over fixed spectral methods. Could further analysis be provided to illustrate specific PDE scenarios where point-calibration particularly excels compared to the fixed approach?
>
> Through further analysis of the experimental results, we find that PCSM particularly performs better over the fixed spectral method (PCSM w/o Cali) in the following specific scenarios:
> - The three fluid-related problems:
>     - Navier-Stokes (relative gain: 22.46%)
>     - Pipe Turbulence (relative gain: 25.76%)
>     - Blood Flow (relative gain: 22.37%)
> - The Plasticity problem (relative gain: 33.88%)
>
> We attribute these superior improvements to the fact that **considering local details is particularly crucial in these scenarios**.
> For fluid dynamics, it is well known that the complexity of fluid behavior involves a significant amount of local detail, especially in turbulent flows.
> In the case of Plasticity, as presented in Figure 3, the local details in the top boundary regions are quite important for correct predictions.
> Therefore, PCSM achieves a marked advantage over the fixed spectral method in these scenarios.
>
>
> ## Question 4
>
> > Q4: Sensitivity to Frequency Parameters: How sensitive is PCSM’s performance to changes in the number of frequency modes (Nk)? Are there guidelines for choosing this parameter in cases where computational resources are limited?
>
> We have included the ablation studies on the frequency number $N^{k}$ in the appendix of our initial submission (Table 8, Section A.4).
> The results show that on Darcy Flow, PCSM with 16 frequencies could even outperform the fixed spectral method (PCSM w/o Cali) with 128 frequencies.
> This indicates that unlike the fixed spectral method (PCSM w/o Cali), **PCSM exhibits lower sensitivity to the number of frequencies**, due to its point-level frequencies augmentation mechanism.
> Consequently, PCSM may offer superior performance in practical scenarios where the availability of a lot of spectral features is constrained by high computational costs.
>
> Regarding the guidelines for choosing the frequency number $N^k$ under limited computational resources, we recommend integrating PCSM with the adaptive frequency selection techniques from prior research [1,2]. Benefiting from the explicit spectral transform operation in PCSM, such integration can be seamlessly achieved.
>
> **Reference**
> - [1] Incremental Spatial and Spectral Learning of Neural Operators for Solving Large-Scale PDEs
> - [2] Adaptive Fourier Neural Operators: Efficient Token Mixers for Transformers
>
>
> ## Question 5
>
> > Q5: Ablation on Multi-Head Mechanism: The multi-head mechanism is mentioned as an enhancement to the spectral mixer. Have the authors conducted any ablation studies to understand how varying the number of heads affects performance?
>
> Thanks for your suggestions about the ablation study on head numbers.
> We complement this ablation on the Darcy Flow problem.
> The results are shown in the table below:
> | Head Number | Relative Error |
> | ---- | ---- |
> | 1    | 4.77e-3 |
> | 2    | 4.79e-3 |
> | 4    | 4.56e-3 |
> | 8    | 4.38e-3 |
> | 16   | 4.54e-3 |
>
> The results indicate that (1) the multi-head mechanism (with 4, 8, and 16 heads) could enhance the performance of operator learning compared to a single head, and (2) PCSM is insensitive to the varying head number. We will incorporate the results in the manuscript.

---

> ### Author Response · Authors · 2024-11-21
> **Response to Reviewer Eei4 [Part 4]**
>
> ## Question 6
>
> > Q6: Explainability of Point-Wise Frequency Preferences: Have the authors considered using interpretability techniques to explain the model’s learned point-wise frequency preferences? For instance, do specific frequencies correspond to particular physical phenomena in the PDE domains?
>
> Thank you for your valuable suggestions about interpretability analysis.
> Regarding the explanation of point-wise frequency preferences, we have conducted two types of analyses in our initial submission (Section 3.4):
>
> 1. We visualize the predicted point-wise frequency preferences across different layers and heads as shown in Figures 6, 7, and 8. Our empirical observations include:
>     - (1) Generally, the model favors low-frequency features for neural operator learning, consistent with previous research emphasizing the importance of lower frequencies.
>     - (2) Boundary regions and areas with rapid changes in physical quantities tend to use more high-frequency spectral features, indicating a higher demand for detailed information in local regions.
>     - (3) The employment of high-frequency spectra is more in the middle layers of the neural network, less in the later layers, and almost negligible in the early layers.
>
> 2. We attempt to transfer the learned frequency preference rules by PCSM (more high-frequency in middle layers) to fixed spectral methods, as shown in Table 5.
> The results indicate that the model taking PCSM’s layer-specific choices (middle layers) achieves better performance.
>
> While we have not yet applied more advanced interpretability techniques to further explain these point-wise frequency preferences, we recognize that it is an intriguing area worthy of further investigation in the future.
>
> ## Question 7
>
> > Q7: Future Extensions of Point-Calibrated Spectral Transform: The paper suggests broader applicability for time-series and computer vision tasks. Could the authors provide some insights into how the Point-Calibrated Spectral Transform might be adapted for non-spectral data, such as images?
>
> Regarding the potential applications of Point-Calibrated Spectral Mechanism (PCSM) in various scenarios, we would like to highlight the following two approaches:
>
> 1. Despite including the spectral transform, we posit that **PCSM could be directly employed for non-spectral data processing** such as images.
> As shown in Remark A.9 (Section A.2.3), we show that PCSM achieves an implicit linear attention mechanism (the neural gates implicitly act as a point-level attention mechanism).
> Therefore, even with limited frequency numbers (low computational cost), like the linear attention mechanism, PCSM could achieve the point-level feature processing required in non-spectral data processing.
>
> 2. While it is beyond the scope of this work, we note that PCSM has the potential to be extended for efficient full-frequency spectral processing of images and videos. The primary challenges of full-frequency spectral processing involve:
>  - (1) **Full-frequency Spectral Feature Calculation**: For images and videos, the Fast Fourier Transform (FFT) can be utilized to compute all spectral features efficiently, due to the regular and uniform properties of these domains. The introduced Point-Calibrated Spectral Transform can be seamlessly integrated into the FFT algorithm, with the calculated spectral gates.
>  - (2) **Full-frequency Spectral Gates Prediction**: The prediction module of spectral gates can be modified for full-frequency gates prediction. For example, for an image of size $s\times s$ (with $N=s \times s$ spatial points), we can predict the spectral gates for each dimension separately and then combine them (via multiplying the gate values of all dimensions) to obtain the gates for specific frequencies. This approach significantly reduces the time complexity compared to predicting the gates for $N$ points directly.
>
> We recognize that there are significant challenges to overcome before PCSM can be applied to general deep learning problems.
> However, we believe that introducing PCSM in general deep learning is promising and warrants further exploration in the future.

---

> ### Author Response · Authors · 2024-11-25
> **Friendly reminder: Have we addressed your concerns?**
>
> Dear Reviewer Eei4,
>
> We hope this message finds you well. We wish to extend our sincere gratitude to you for the expeditious and meticulous review you performed on our work.
>
> As we draw closer to the end of the discussion phase, we want to confirm that we have adequately addressed all of your concerns and questions about our work. We would be grateful if you could provide us with your further thoughts on our detailed rebuttal.
>
> If there are any additional questions or concerns that we have not yet fully addressed for a better score, please feel free to inform us. We will be happy to answer them as your feedback is crucial to us.
>
> Best regards,
>
> PCSM Authors.

---

> ### Author Response · Authors · 2024-12-01
> **Second Friendly Reminder: Awaiting Your Feedback on Our Rebuttal**
>
> Dear Reviewer Eei4,
>
> We hope this message finds you well. As we approach the delayed end of the discussion period, We would like to extend our gratitude again for your review of this work.
>
> We have submitted a rebuttal to address the points raised in your review and have been looking forward to your further thoughts. Could you please confirm whether you have had the opportunity to review our responses? We are eager to ensure that all your concerns have been adequately addressed.
>
> Thank you very much for your time and consideration.
>
> Best regards,
>
> PCSM Authors.

---

> > ### Comment · Reviewer_Eei4 · 2024-12-02
> > **Keep the same score**
> >
> > Thanks for your detailed response to my reviews. There are still a few concerns.
> >
> > - The authors assert that PCSM is not sensitive to the number of frequencies or the head counts, which seems counterintuitive. Meanwhile, the experimental results provided are insufficient to support this claim.
> > - Regarding real-world or industry-related scenarios, I am not convinced that the current experimental results are relative enough.
> > - Regarding interpretability, the visualizations and explanations are somewhat weak. Utilizing PCSM's layer selection with fixed spectral methods appears to intuitively enhance these methods. This approach, however, contradicts the claim that PCSM is insensitive to the number of frequencies.
> >
> > That said, I acknowledge that the paper makes a valuable contribution to tackling PDE problems through the innovative Point-Calibrated Spectral Neural Operator, which combines the flexibility of attention-based methods with the spectral continuity constraints inherent in spectral-based neural operators.
> >
> > I intend to maintain the same score based on the analysis provided above.

---

> > > ### Author Response · Authors · 2024-12-02
> > > **Clarification on the influence of frequency counts**
> > >
> > > Thank you very much for your response to our rebuttal and your recognition of the contribution of this work.
> > > Your valuable comments have greatly enhanced this work.
> > >
> > > First, we wish to rectify some possible misunderstandings about the influence of frequency numbers in the following:
> > >
> > > Our statement was not that "**PCSM performance is unaffected by the number of frequencies**," but rather that "**PCSM maintains good performance even with a smaller number of frequencies compared to the fixed spectral method (PCSM w/o Cali)**". In other words, PCSM exhibits less dependency on the frequency count compared to the fixed spectral method.
> > > We believe the results provided in Table 8 sufficiently lead to this conclusion.
> > > For convenience, we place the results of Table 8 below:
> > >
> > > | Problems          | Models        | 16 Frequencies | 32 Frequencies | 64 Frequencies | 128 Frequencies |
> > > |------------------|---------------|----------------|----------------|----------------|-----------------|
> > > | Darcy Flow       | PCSM (w/o Cali) | 1.04e-2      | 8.08e-3        | 6.15e-3        | 5.31e-3         |
> > > | Darcy Flow       | PCSM          | 5.10e-3      | 5.02e-3        | 4.85e-3        | 4.38e-3         |
> > > | Navier-Stokes    | PCSM (w/o Cali) | 1.18e-1      | 1.05e-1        | 9.47e-2        | 8.37e-2         |
> > > | Navier-Stokes    | PCSM          | 9.51e-2      | 8.77e-2        | 7.44e-2        | 7.28e-2         |
> > >
> > > From the table, it can be observed that as the number of frequencies decreases, the performance of the fixed spectral method (PCSM w/o Cali) drops more rapidly, whereas PCSM's performance degrades at a slower rate. Moreover, even using fewer frequencies, PCSM outperforms "PCSM w/o Cali" with more frequencies.
> > > Additionally, increasing the number of frequencies could improve PCSM's performance.
> > > Therefore, in scenarios where obtaining sufficient frequency features is computationally expensive, PCSM could still performs well with fewer frequencies compared to the fixed spectral method by utilizing the learned frequency preference.

---

> > > ### Author Response · Authors · 2024-12-02
> > > **Response to remaining concerns**
> > >
> > > Below is our response to your remaining concerns:
> > >
> > > **Concern 1**
> > >
> > > We note that the influences of frequency numbers and head numbers are different, both of which could be supported by corresponding results.
> > >
> > > (1) Regarding the impact of the frequency number, as the complemented clarification, PCSM demonstrates a reduced dependency on the large frequency number compared to the fixed spectral method, rather than the "counterintuitive conclusion" that PCSM performance is unaffected by frequency number. This is validated by the ablation of the frequency numbers.
> > >
> > > (2) Regarding the head number, our complemented results (in response to your question 5) indicate that PCSM's performance does not significantly change across different numbers of heads. The standard deviation in performance between 4, 8, and 16 heads is only `9.87e-5`.
> > >
> > > **Concern 2**
> > >
> > > Following previous works [1,2,3,4,5,6], we believe current experimental results (including both structured and unstructured problems, and covering some realistic scenarios) could sufficiently demonstrate the effectiveness of PCSM across various scenarios.
> > >
> > > (1) Regarding the industry-related synthesized problems, as mentioned in the initial rebuttal, we have included problems from real-world or industry-related scenarios (such as Composite from the aerospace industry and Blood Flow from a practical medical scenario). Incorporating additional industry-related synthesized problems could not significantly extend the current experimental scope nor substantially alter existing conclusions. Therefore, we believe that the current set of experimental problems adequately demonstrates that PCSM is effective across a variety of scenarios. Nevertheless, we are committed to exploring more industry-related scenarios in the future.
> > >
> > > (2) Regarding more realistic scenarios, we understand your preference for "demonstrating PCSM’s utility outside synthetic PDE problems". It is important to note, however, that genuine industrial data is extremely costly and typically not publicly available, making it challenging to incorporate into scientific experiments. Previous works, including FNO [1], Geo-FNO [2], U-NO [3], Galerkin [4], GNOT [5], Transolver [6], etc., have not featured experiments with such authentic datasets.
> > >
> > > **Concern 3**
> > >
> > > We note that the provided visualizations and explanations could validate that PCSM effectively learns reasonable frequency preference, the focus of this work, although they are somewhat weak in terms of "interpretability analysis" (e.g. finding specific frequencies correspond to particular physical phenomena).
> > >
> > > (1) While professional interpretability analysis is attractive, they are beyond the central scope (integrating the flexibility of attention models and constraints of spectral models) of this work. They could not significantly contribute to the core conclusion of this work. However, we will explore relevant techniques in the future.
> > >
> > > (2) In addition, regarding your mentioned "contradiction", as the complemented clarification, we did not imply that PCSM's performance is independent of the number of frequencies. Instead, we demonstrated that PCSM can maintain good performance even with fewer frequency features by leveraging learned point-level frequency preferences. The learned frequency preference (more high-frequency or low-frequency features in different layers) could instruct the spectral frequency design in fixed spectral methods. Thus, there is no contradiction between the conclusion on the number of frequencies (as discussed in the clarification) and the results from the fixed frequency design (presented in Section 3.4).
> > >
> > > **The Contribution of This Work**
> > >
> > > As you recognized, this work makes a valuable contribution to solving PDE problems through the innovative Point-Calibrated Spectral Neural Operator. Additionally, we have:
> > > 1. Demonstrated its significance across a wide range of problems (Sections 3.1, 3.2, and 3.3).
> > > 2. Provided an intuitive analysis of the learned frequency preferences (Section 3.4).
> > > 3. Strengthened the theoretical foundations by explaining how PCSM learns integral neural operators (Theorem A.8, Section A.2.3) and how it integrates the integral kernels of spectral models and linear attentions (Remark A.9, Section A.2.3).
> > >
> > > Overall, we believe that the innovative method, along with the extensive experiments, intuitive explanations, and theoretical foundations, clearly indicate the significant contribution of this work to the field. We sincerely hope that you will reconsider your assessment in light of these clarifications.
> > >
> > > **Reference**
> > > - [1] Fourier Neural Operator for Parametric Partial Differential Equations
> > > - [2] Fourier Neural Operator with Learned Deformations for PDEs on General Geometries
> > > - [3] U-NO: U-shaped Neural Operators
> > > - [4] Choose a Transformer: Fourier or Galerkin
> > > - [5] GNOT: A General Neural Operator Transformer for Operator Learning
> > > - [6] Transolver: A Fast Transformer Solver for PDEs on General Geometries

---

### Meta-Review · Area_Chair_QaL7 · 2024-12-22

**Metareview:**

Summary.
The paper introduces the Point-Calibrated Spectral Mixer (PCSM) Neural Operator. This new approach combines the flexibility of attention-based methods and the spectral continuity constraints from spectral-based neural operators.

Strengths.
This paper introduces a novel hybrid approach by integrating spectral methods with point-wise adaptability, which is an advancement in the field of neural PDE solvers.
The experiments cover a range of structured and unstructured PDE problems, offering quantitative and qualitative results that consistently show the advantages of PCSM over baselines.

Weaknesses.
It is unclear how sensitive is the proposed method to changes in the number of frequency modes. The paper did not provide convincing guidelines for choosing this parameter in cases where computational resources are limited.
The authors did not provide a detailed explanation of how low frequency and high frequency are defined, as well as their respective roles.
The introduction to neural operators in general and point-calibrated spectral neural operators is short and feels somewhat cursory.

Missing.

The paper is missing a thorough discussion on computational complexity of different operations in the proposed method.
A clear explanation of "Pointwise Frequency Preference Learning" to support the claims of frequency selection and number of frequency heads is missing.

Reasons.
The following factors contributed the most to my decision.
The proposed method borrows significantly from the existing closely related work.
The central contribution is the PCSM, which offers better performance, but the claims about point-wise frequency selection require further discussion and evidence.

**Additional Comments On Reviewer Discussion:**

The paper had some discussion between authors and reviewers.

Reviewers main questions were about computational complexity, choice of frequency modes, novelty of the proposed method over existing work on neural operators, and explanation of point-wise frequency preference learning.

Authors provided detailed responses with clarifications and additional experiments.

Reviewers concerns largely remain unresolved during rebuttal period.
One reviewer summarized that "The authors assert that PCSM is not sensitive to the number of frequencies or the head counts, which seems counterintuitive. Meanwhile, the experimental results provided are insufficient to support this claim."
One reviewer argued that "I would not buy the explanation of "Pointwise Frequency Preference Learning" claimed by the authors."

---

### Decision · Program_Chairs · 2025-01-22

Reject